# MAKING LLaMA SEE AND DRAW WITH SEED TOKENIZER

**Yuying Ge**[1*]**, Sijie Zhao**[1*]**, Ziyun Zeng**[2]**, Yixiao Ge**[1,2†]**, Chen Li**[2]**, Xintao Wang**[1,2]**, Ying Shan**[1,2]

[1]Tencent AI Lab, [2]ARC Lab, Tencent PCG

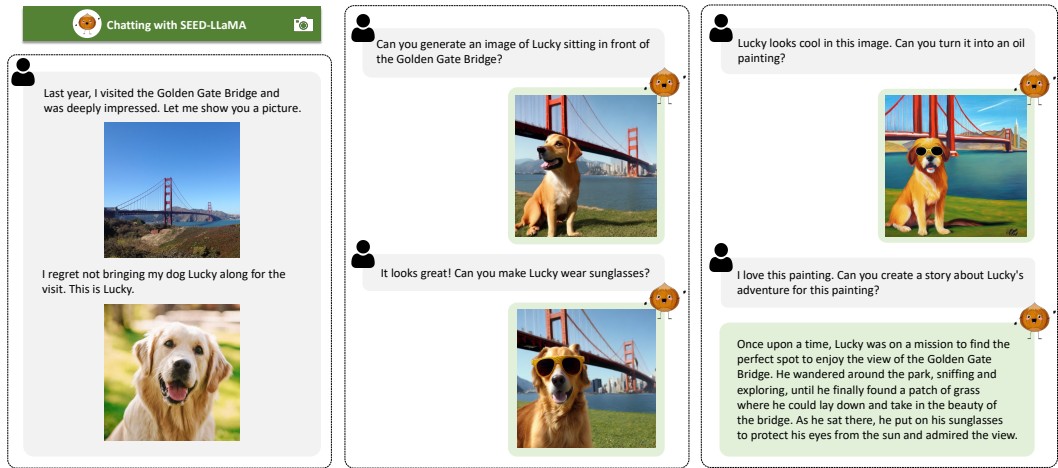

Figure 1: The introduced SEED-LLaMA, a multimodal AI assistant, demonstrates **emergent ability** in the multi-turn in-context image and text generation given multimodal instructions.

## ABSTRACT

The great success of Large Language Models (LLMs) has expanded the potential of multimodality, contributing to the gradual evolution of General Artificial Intelligence (AGI). A true AGI agent should not only possess the capability to perform predefined multi-tasks but also exhibit emergent abilities in an open-world context. However, despite the considerable advancements made by recent multimodal LLMs, they still fall short in effectively unifying comprehension and generation tasks, let alone open-world emergent abilities. We contend that the key to overcoming the present impasse lies in enabling text and images to be represented and processed interchangeably within a unified autoregressive Transformer. To this end, we introduce **SEED**, an elaborate image tokenizer that empowers LLMs with the ability to **SEE** and **D**raw at the same time. We identify two crucial design principles: (1) Image tokens should be independent of 2D physical patch positions and instead be produced with a *1D causal dependency*, exhibiting intrinsic interdependence that aligns with the left-to-right autoregressive prediction mechanism in LLMs. (2) Image tokens should capture *high-level semantics* consistent with the degree of semantic abstraction in words, and be optimized for both discriminativeness and reconstruction during the tokenizer training phase. With SEED tokens, LLM is able to perform scalable multimodal autoregression under its original training recipe, i.e., next-word prediction. SEED-LLaMA is therefore produced by large-scale pretraining and instruction tuning on the interleaved textual and visual data, demonstrating impressive performance on a broad range of multimodal comprehension and generation tasks. More importantly, SEED-LLaMA has exhibited compositional emergent abilities such as multi-turn in-context multimodal generation, acting like your AI assistant. The code (training and inference) and models are released in `https://github.com/AILab-CVC/SEED`.

---

[*]Equal Contribution.

[†]Correspondence to `yixiaoge@tencent.com`.

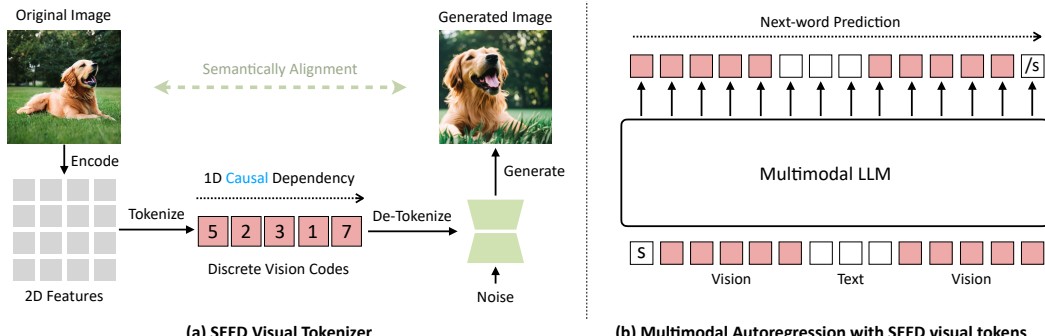

Figure 2: (a) SEED is a discrete image tokenizer, producing quantized visual codes with 1D causal dependency and high-level semantics. (b) With SEED tokenizer, LLM is able to perform scalable multimodal autoregression on interleaved visual and textual data with next-word-prediction objective.

# 1 INTRODUCTION

In recent years, Large Language Models (Touvron et al., 2023; Brown et al., 2020; Chowdhery et al., 2022) (LLMs) pre-trained on massive text corpus with straightforward training objectives such as next-word prediction have exhibited remarkable abilities to understand, reason, and generate texts across a variety of open-ended tasks. Recent studies further exploit the strong generality of LLMs to improve visual understanding or generation tasks, collectively referred to as Multimodal LLM (MLLM). While these studies have contributed to technological advancements, MLLMs have yet to achieve the remarkable success of LLMs in terms of emergent capabilities. We have made a bold assumption that the premise for the emergence of multimodal capabilities is that text and images can be represented and processed **interchangeably** in a unified autoregressive Transformer.

We posit that a proper visual tokenizer is the key as it can facilitate the follow-up multimodal training by (i) easing the semantic alignment between visual and word tokens, and (ii) enabling LLM's original training recipe (i.e., next-word prediction) for multimodal data without specific adaptation for visual tokens. Representing images as a sequence of discrete IDs is naturally compatible with the autoregressive training objective of LLMs. But unfortunately, works (Ramesh et al., 2021; Ding et al., 2021) that utilize discretized visual tokens for multimodal tasks have receded from prominence, as such models generally rely on super-scale training to converge, leading to substantial training costs. Moreover, we empirically found that the dominant tokenizer VQ-VAE (Van Den Oord et al., 2017) in existing works captures too low-level information for LLMs to effectively perform multimodal comprehension tasks. Existing image tokenizers fail to meet the requirements of unifying the generation of images and texts and facilitating multimodal training.

To this end, we introduce **SEED**, a VQ-based image tokenizer that produces discrete visual codes with 1D causal dependency and necessary high-level semantics for both visual comprehension and generation tasks, as shown in Fig. 2 (a). The off-the-shelf LLMs can be readily equipped with SEED by treating discrete visual tokens as new words and updating the vocabulary. We would like to emphasize the design principles of SEED. (1) *Why causal-dependent tokens?* Existing visual tokens (*e.g.*, from VQ-VAE or CLIP-ViT (Sun et al., 2023a)) are generated using 2D context, which is incompatible with the unidirectional attention in dominant LLMs and counterintuitive for text-to-image tasks requiring raster order prediction. Thus, we convert 2D raster-ordered embeddings into a sequence of semantic codes with 1D causal dependency. (2) *Why high-level semantics?* Since visual and textual tokens in LLMs are expected to be interoperable—sharing weights and training objectives—they should encompass the same degree of semantics to prevent misalignment, i.e., the high-level semantics inherently present in words.

Specifically, the SEED tokenizer is composed of a ViT encoder (Dosovitskiy et al., 2020), Causal Q-Former, VQ Codebook (Van Den Oord et al., 2017), multi-layer perceptron (MLP), and a UNet decoder (Ronneberger et al., 2015). The ViT encoder and UNet decoder are directly derived from the pre-trained BLIP-2 (Li et al., 2023c) and unCLIP-SD model (Rombach et al., 2022; Ramesh et al., 2022), respectively. (1) *Tokenize:* Causal Q-Former converts 2D raster-ordered features produced by the ViT encoder into a sequence of causal semantic embeddings, which are further discretized by the VQ Codebook. (2) *De-Tokenize:* The discrete visual codes are decoded into generation embedding

via MLP. The generation embedding is aligned with the latent space of unCLIP-SD so that realistic images with consistent semantics can be generated using the off-the-shelf SD-UNet.

We further present **SEED-LLaMA** by equipping the pre-trained LLM (Touvron et al., 2023) with SEED tokenizer. SEED-LLaMA is pretrained on multimodal data, including image-text pairs, video-text pairs, and interleaved image-text data, toward the training objective of next-word prediction as shown in Fig. 2 (b). Such an easy-to-implement and unified proxy task facilitates scalable multimodal pretraining. We further apply multimodal instruction tuning to align SEED-LLaMA with human instructions through supervised fine-tuning. Our model demonstrates extensive emergent abilities such as multi-turn in-context image and text generation given multimodal instructions as shown in Fig. 1. We also benchmark on a broad range of tasks including image captioning, image/video question answering, and text-to-image generation, receiving competitive performance.

In summary, our contributions are three-fold. (1) We introduce SEED, an advanced image tokenizer, designed based on the insights that visual tokens compatible with LLMs should capture high-level semantics while being generated with 1D causal dependency. The tailored SEED improves the scalability of subsequent multimodal training. (2) We present SEED-LLaMA, composed of a pre-trained LLM and SEED tokenizer, through large-scale multimodal pretraining and instruction tuning under the next-word-prediction training objective. It successfully unified multimodal comprehension and generation tasks in one framework. (3) SEED-LLaMA shows competitive results on existing multimodal tasks (e.g., text-to-image, image-to-text) and further demonstrates emergent abilities in multi-turn in-context multimodal understanding, reasoning, and generation.

## 2 RELATED WORK

**MLLMs for Comprehension and Generation.** With the impressive success of Large language models (Touvron et al., 2023; Brown et al., 2020; Chowdhery et al., 2022) (LLMs), recent studies work on Multimodal LLM (MLLM) to improve visual **comprehension** through utilizing the strong generality of LLMs. Previous work (Ye et al., 2023; Li et al., 2023c; Zhu et al., 2023a; Zhang et al., 2023b; Gao et al., 2023; Liu et al., 2023b; Alayrac et al., 2022; Driess et al., 2023) align visual features of pre-trained image encoder with LLMs on image-text datasets. However, these work commonly use the prediction of the next *text token* as the objective, thus can only output texts.

To empower LLMs with the image **generation** ability, CogView (Ding et al., 2021) pre-trains a visual tokenizer by reconstructing image pixels, and fine-tunes GPT (Brown et al., 2020) with the objective of next-token prediction. GILL (Koh et al., 2023a) learns a mapping between the embeddings of a LLM and a frozen text-to-image generation model. Both work aim to generate images with LLMs, without being explicitly designed for unifying multimodal comprehension and generation.

Our concurrent works (Sun et al., 2023b; Lili et al., 2023) both perform multimodal autoregression including the generation of images and texts. CM3Leon (Lili et al., 2023) utilizes discrete visual codes from a image tokenizer (Gafni et al., 2022) pre-trained on image pixel reconstruction and performs image-to-text and text-to-image autoregression. However, it yields suboptimal performance in visual comprehension tasks (e.g., CIDEr 61.6 vs. ours 126.9 on COCO image captioning) because the image tokenizer captures too low-level information. Emu (Sun et al., 2023b) employs continuous visual representations and is pre-trained on interleaved multimodal sequences through classifying the next text token or **regressing** the next visual embedding. For image generation, Emu further fine-tunes a SD model to accommodate the output representations from the LLM. By contrast, we pre-train a discrete image tokenizer, where the visual codes can be decoded to realistic images using the off-the-shelf SD model, and perform multimodal autoregressive with a unified next-word-prediction objective, which facilitates scalable multimodal training.

**Visual Tokenizer.** Visual tokenizer aims to represent images as a sequence of discrete tokens. Previous work (Van Den Oord et al., 2017; Ramesh et al., 2021; Esser et al., 2021; Gu et al., 2022) trains a Vector Quantized Variational AutoEncoders (VQ-VAE) by reconstructing image pixels, which captures only low-level details such as color, texture and edge. Beit v2 (Peng et al., 2022) trains a visual tokenizer through reconstructing high-level features from the teacher model, but its visual codes from 2D features of a vision transformer (Dosovitskiy et al., 2020) are incompatible with the unidirectional attention in dominant LLMs for image generation. By contrast, we present SEED tokenizer, which produces discrete visual codes with 1D causal dependency and high-level semantics.

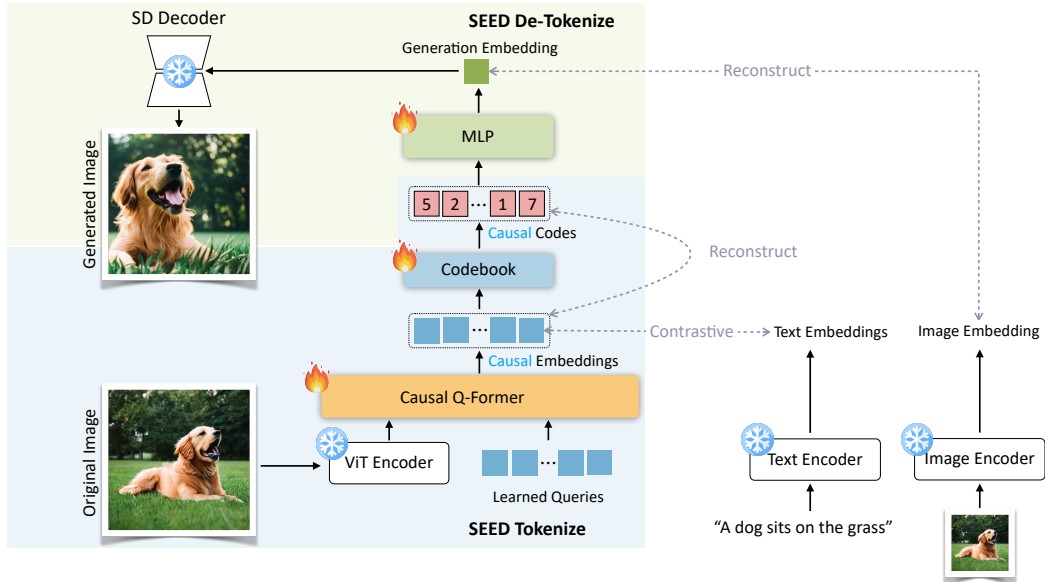

Figure 3: Overview of **SEED** tokenizer, which produces discrete visual codes with causal dependency and high-level semantics. The generation embedding from visual codes can be decoded to realistic images with the frozen unCLIP (Ramesh et al., 2022) SD, which is conditioned on image embedding.

## 3  METHOD

### 3.1  SEED TOKENIZER

As shown in Fig. 3, the SEED tokenizer consists of the following components for tokenization and de-tokenization:

(1) ViT encoder (Dosovitskiy et al., 2020), which is derived from the pre-trained BLIP-2 (Li et al., 2023c) and is frozen, providing visual features as the inputs of the Causal Q-Former.

(2) Causal Q-Former, which is trained (Enq. 1) to convert 2D raster-ordered features ($16 \times 16$ tokens) produced by the ViT encoder into a sequence of causal embeddings (32 tokens).

(3) VQ Codebook (Van Den Oord et al., 2017), which is trained (Enq. 2) to discretize the causal embeddings from Causal Q-Former to quantized visual codes (32 tokens) with causal dependency.

(4) Multi-layer perceptron (MLP), which is trained (Enq. 3) to decode the visual codes from VQ Codebook into generation embedding (1 token) for the alignment with the latent space of the pre-trained unCLIP-SD (Ramesh et al., 2022) conditioned on image embedding.

(5) UNet decoder (Ronneberger et al., 2015), which is derived from the unCLIP-SD to encode realistic images from the generation embedding, and is kept frozen.

We pre-train SEED tokenizer on image-text pairs including CC3M (Sharma et al., 2018), Unsplash (Luke Chesser, 2023), LAION-COCO (Christoph et al., 2022) and MS-COCO (Chen et al., 2015).

#### 3.1.1  TRAINING STAGE I: CAUSAL Q-FORMER

As shown in Fig. 3, we first train Causal Q-former to extract a fixed number of output features with causal dependency from from the image encoder through contrastive learning. Specifically, a set number of learnable query embeddings (32 tokens) and features of a pre-trained ViT encoder (Sun et al., 2023a) are fed into the Causal Q-former to encode a fixed number of causal embeddings (32 tokens) of the input image. The query embeddings can interact with only previous queries through self-attention layers with causal mask, and interact with frozen image features through cross-attention layers. We adopt contrastive learning to maximize the similarity between the **final** causal embedding $\hat{c}$ and text features $t$ of the corresponding caption, while minimizing the similarity between the **final** causal embedding and text features of other captions in a batch, using Noise-Contrastive Estimation Oord et al. (2018) as below,

$$\mathcal{L}_{\text{CQ}} = -\log \frac{\exp(\hat{c}_i^T t_i / \tau)}{\sum_{j=1}^{B} \exp(\hat{c}_i^T t_j / \tau)} - \log \frac{\exp(t_i^T \hat{c}_i / \tau)}{\sum_{j=1}^{B} \exp(t_i^T \hat{c}_j / \tau)} \tag{1}$$

where $B$ is the number of the batch size and $\tau$ is the learnable temperature parameter.

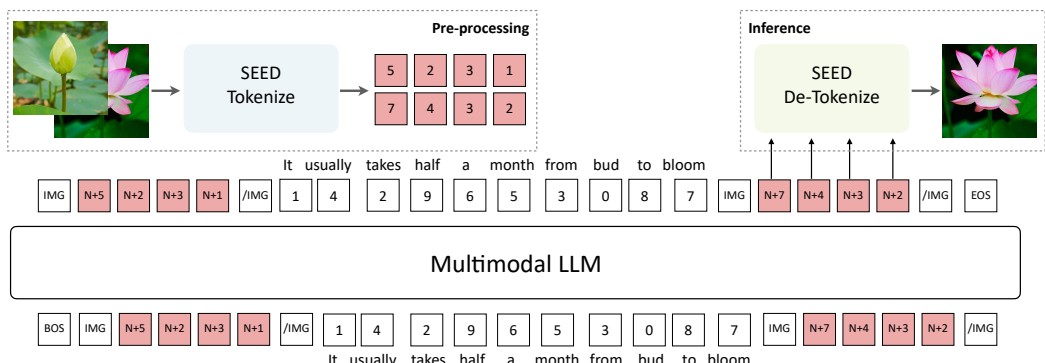

Figure 4: Overview of the multimodal autoregressive pretraining on interleaved visual and textual data for **SEED-LLaMA**. Visual inputs are pre-processed into discrete tokens to conserve computational resources. Given the multimodal discrete sequence, a unified next-word-prediction objective is employed. During inference, visual codes are decoded into a realistic image by SEED De-Tokenization.

### 3.1.2 TRAINING STAGE II: VISUAL TOKENIZE AND DE-TOKENIZE

As shown in Fig. 3, we train a VQ codebook to discretize the causal embeddings (32 tokens) into quantized visual codes (32 tokens) for visual tokenization, and a MLP to decode the visual codes into generation embedding (1 token) for visual de-tokenization. The vocabulary size of the VQ codebook is set as 8192. Specifically, causal embeddings $C$ from the frozen Causal Q-Former are first fed into a linear projection layer to reduce the dimensions as $C'$ for improving codebook utilization following previous work Peng et al. (2022). A quantizer looks up the nearest neighbor embeddings $Z$ in the codebook for $C'$ and obtains the corresponding codes. We employ a decoder, which is a multi-layer Transformer (Dosovitskiy et al., 2020), to reconstruct the continuous causal embeddings $C$ from discrete codes as $C^{\text{recon}}$. We employ a MLP to reconstruct the image embedding $I$ (1 token) of a frozen unCLIP-SD from discrete codes as generation embedding $G$. As shown in Enq. 2, during training, besides the original VQ objective Van Den Oord et al. (2017) (the first and second term), we maximize the cosine similarity between reconstructed causal embeddings $C^{\text{recon}}$ and causal embeddings $C$ (the third term), and minimize the MSE loss between generation embedding $G$ and image embedding $I$ of unCLIP-SD (the fourth term).

$$\mathcal{L}_{\text{VQ}} = ||\text{sg}[C'] - Z||_2^2 + ||C' - \text{sg}[Z]||_2^2 - \cos(C^{\text{recon}}, C) + \mathcal{L}_{\text{MLP}} \tag{2}$$

$$\mathcal{L}_{\text{MLP}} = \text{MSE}(G, I) \tag{3}$$

where sg[.]denotes the stop-gradient operation. During inference, the generation embedding $G$ are fed into the off-the-shelf SD-UNet to decode realistic images.

## 3.2 SEED-LLaMA

### 3.2.1 TRAINING STAGE I: MULTIMODAL PRETRAINING

As shown in Fig. 4, SEED-LLaMA adopts a unified next-word-prediction training objective on interleaved visual and textual data. Specifically, visual inputs are first discretized into a sequence of causal codes by SEED tokenizer. Then the interleaved visual codes and text tokens are fed into the pretrained LLM for performing multimodal autoregression, where the visual codes are treated as new words and the vocabulary of the LLM is updated accordingly. We maximize the likelihood in a unified autoregressive manner as follows:

$$L(\mathcal{U}) = \sum_i \log P\left(u_i \mid u_{i-k}, \ldots, u_{i-1}; \Theta\right) \tag{4}$$

where $u_i$ represents visual code or text token, and $\Theta$ denotes the the parameters of the transformer. We initialize SEED-LLaMA from a pre-trained LLM, and add 8192 visual codes to the vocabulary. The embedding layer and decoder head layer in the transformer are expanded and the parameters of added visual codes are randomly initialized.

For efficiency, we first train SEED-LLaMA using LoRA (Hu et al., 2021) tuning and together optimize the parameters of the embedding layer and decoder head layer due to the added visual codes. We

Table 1: Evaluation of Image-Text Retrieval. Causal codes are quantized causal embeddings.

| Model | Flickr30K (1K test set) | | | | | | | COCO (5K test set) | | | | | | |
| | Image → Text | | | Text → Image | | | | Image → Text | | | Text → Image | | | |
| | R@1 | R@5 | R@10 | R@1 | R@5 | R@10 | R@m | R@1 | R@5 | R@10 | R@1 | R@5 | R@10 | R@m |
|---|---|---|---|---|---|---|---|---|---|---|---|---|---|---|
| BLIP-2 (Li et al., 2023c) | 81.9 | 98.4 | 99.7 | **82.4** | **96.5** | **98.4** | 92.9 | 65.3 | 89.9 | 95.3 | **59.1** | 82.7 | **89.4** | 80.3 |
| SEED (causal embedding) | **91.0** | **99.5** | **100.0** | 79.3 | 94.8 | 97.1 | **93.6** | **74.2** | **93.1** | **96.7** | 59.0 | **82.8** | 89.2 | **82.5** |
| SEED (causal code) | 85.4 | 98.3 | 99.6 | 73.7 | 92.3 | 95.7 | 90.8 | 66.9 | 89.3 | 94.4 | 53.2 | 78.8 | 86.6 | 78.2 |

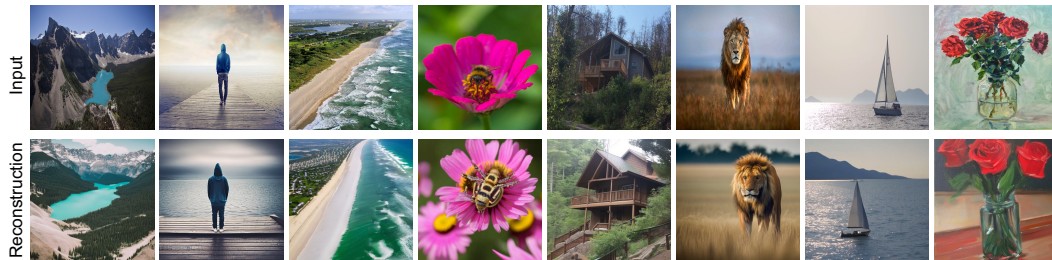

Figure 5: Reconstruction images of SEED tokenizer (*i.e.*, original image → SEED tokenize → causal visual codes → SEED de-tokenize → reconstructed image).

then merge the parameters of LoRA onto the LLM backbone and fine-tune all parameters except for the embedding layer. We freeze the embedding layer since we observe that fine-tuning it together with other parameters can lead to unstable training loss, which is also reported in BLOOM (Scao et al., 2022) and GLM-130B (Zeng et al., 2022). We preprocess the images and videos into discrete tokens beforehand to conserve computational resources. We perform pretraining using two versions of LLM, Vicuna-7B and Llama2-chat-13B, with 64 A100-40G GPUs, and yield SEED-LLaMA-8B (144 hours) and SEED-LLaMA-14B (216 hours), respectively. See Appendix. B for details.

### 3.2.2 TRAINING STAGE II: MULTIMODAL INSTRUCTION TUNING

We perform multimodal instruction tuning on SEED-LLaMA to align it with human instructions through supervised finetuning on public datasets. The details of datasets can be found in Appendix. C. We fine-tune a LoRA module on the pre-trained SEED-LLaMA with the template as below,

$$\text{USER: <Instruction> ASSISTANT: <Answer>} \tag{5}$$

Only the content of <Answer> is accounted for loss. The overall instruction tuning phase takes 16 hours for SEED-LLaMA-8B and 27 hours for SEED-LLaMA-14B with 32 A100-80G GPUs.

### 3.3 SEED TOKENIZER

## 4 EXPERIMENT

**Evaluation of Causal Embeddings.** We evaluate the performance of Causal Q-Former on the image-text retrieval using COCO (Lin et al., 2014) and Flickr30K (Young et al., 2014). The performance is measured by *Recall@K* (R@K). Note that we adopt the dual-stream paradigm for inference and remove the image-text-matching (ITM) re-rank module in BLIP-2 for a fair comparison. As shown in Tab. 1, our Causal Q-former achieves better results than BLIP-2 in terms of an aggregated metric *Recall@mean*. It demonstrates that the output query embeddings with causal dependency do not drop performance than the output embeddings with bi-directional attention in BLIP-2.

**Evaluation of Causal Codes.** We evaluate causal codes on the image-text retrieval, where the reconstructed embeddings from causal codes are used for retrieval. As shown in Tab. 1, discrete codes exhibit competitive performance compared to BLIP-2, which demonstrates that the discrete codes from SEED tokenizer capture high-level semantics, which are suitable for visual comprehension.

We visualize the reconstructed images of SEED tokenizer in Fig. 5. Through obtaining the generation embedding from the causal visual codes, realistic images can be generated using the frozen SD-UNet, which maintain consistent semantics with inputs. *The above evaluation and visualization demonstrate the versatility of SEED visual tokens for both comprehension and generation tasks.*

Table 2: Comparison for multimodal comprehension. "Image Gen" denotes whether the model can generate images besides texts, and "-I" denotes the instruction tuned model. The best results are **bold** and the second best are underlined.

| Models | Size | Image Gen | Image-Text Tasks | | | | | Video-Text Tasks | | |
|---|---|---|---|---|---|---|---|---|---|---|
| | | | COCO | VQAv2 | OKVQA | VizWiz | SEED Bench | MSVDQA | MSRVTTQA | NExTQA |
| Flamingo (Alayrac et al., 2022) | 9B | × | 79.4 | 51.8 | 44.7 | 28.8 | 42.7 | 30.2 | 13.7 | 23.0 |
| BLIP-2 (Li et al., 2022) | 4.1B | × | **144.5** | 63.0 | 40.7 | 29.8 | 49.7 | 33.7 | 16.2 | - |
| InstructBLIP (Li et al., 2023c) | 8.1B | × | - | - | - | 34.5 | **58.8** | 41.8 | 22.1 | - |
| Kosmos-1 (Huang et al., 2023) | 1.6B | × | 84.7 | 51.0 | - | 29.2 | - | - | - | - |
| Kosmos-2 (Peng et al., 2023) | 1.6B | × | - | 45.6 | - | - | 54.4 | - | - | - |
| MetaLM (Hao et al., 2022) | 1.7B | × | 82.2 | 41.1 | 11.4 | - | - | - | - | - |
| IDEFICS (Laurençon et al., 2023) | 80B | × | 91.8 | 60.0 | 45.2 | 36.0 | - | - | - | - |
| IDEFICS-I (Laurençon et al., 2023) | 80B | × | 117.2 | 37.4 | 36.9 | 26.2 | 53.2 | - | - | - |
| CM3Leon (Lili et al., 2023) | 7B | ✓ | 61.6 | 47.6 | 23.8 | 37.6 | - | - | - | - |
| Emu (Sun et al., 2023b) | 14B | ✓ | 112.4 | 52.0 | 38.2 | 34.2 | 47.3 | 18.8 | 8.3 | 19.6 |
| Emu-I (Sun et al., 2023b) | 14B | × | 117.7 | 40.0 | 34.7 | 35.4 | 58.0 | 32.4 | 14.0 | 6.8 |
| **SEED-LLaMA** | 8B | ✓ | 123.6 | 44.2 | 29.2 | 21.5 | 42.2 | 11.5 | 5.0 | 14.3 |
| **SEED-LLaMA-I** | 8B | ✓ | 124.5 | **66.2** | **45.9** | **55.1** | 51.5 | 40.9 | 30.8 | **24.9** |
| **SEED-LLaMA** | 14B | ✓ | 125.0 | 48.1 | 27.1 | 23.3 | 46.0 | 13.9 | 3.7 | 11.3 |
| **SEED-LLaMA-I** | 14B | ✓ | 126.9 | 63.4 | 43.2 | 49.4 | 53.7 | **45.2** | **35.3** | 24.7 |

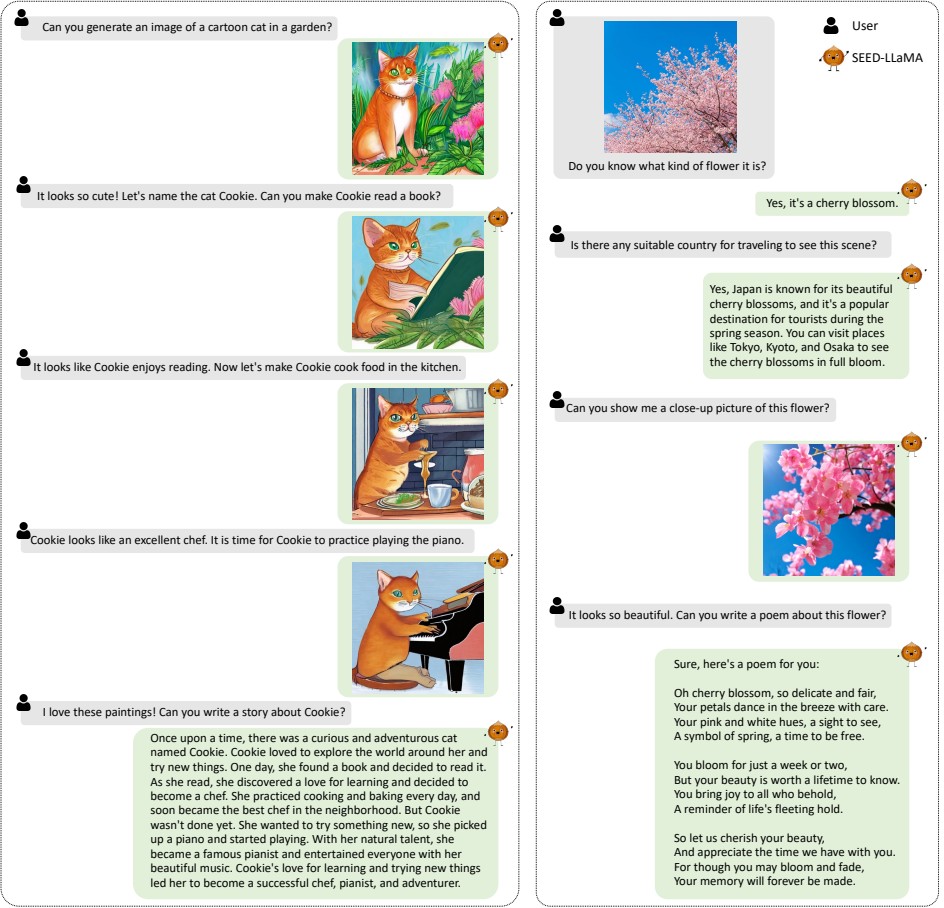

Figure 6: Qualitative examples of multi-turn in-context image and text generation by SEED-LLaMA given multimodal instructions.

## 4.1 SEED-LLAMA

### 4.1.1 QUANTITATIVE EVALUATION

**Multimodal Comprehension.** We evaluate SEED-LLaMA on a wide range of multimodal comprehension tasks including image captioning and image/video question answering. Details of these

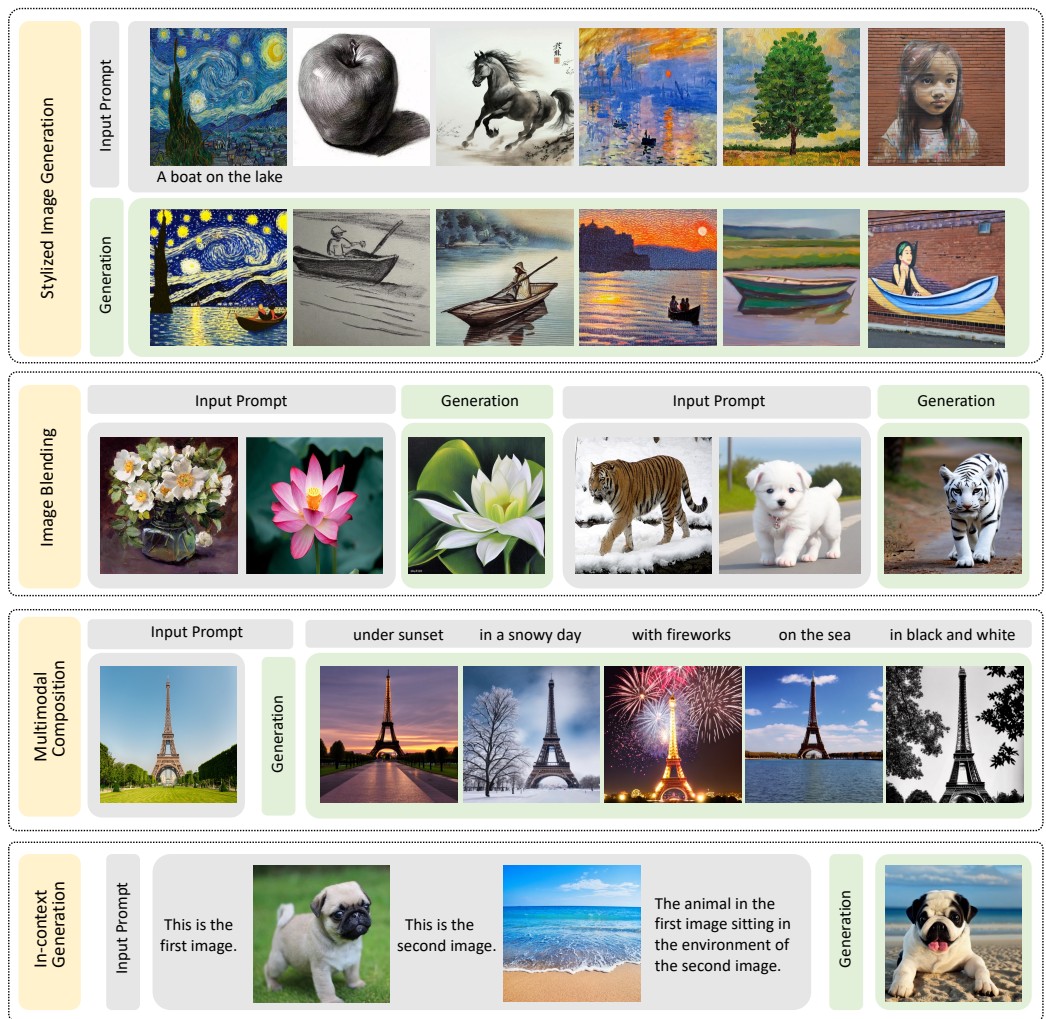

Figure 7: Qualitative examples of compositional image generation by SEED-LLaMA.

benchmarks and evaluation metrics are provided in Appendix. D. As shown in Tab. 2, our SEED-LLaMA achieves competitive performance in both the image and video understanding tasks compared with MLLMs that use continuous visual representations. The results demonstrate that our SEED tokenizer can generate discrete visual codes with high-level semantics, which facilities the visual comprehension. We can observe that pretraining from a LLM with larger model size improves performance on SEED-Bench and instruction tuning further contributes to enhanced results. Note that as pointed out by recent work (Liu et al., 2023c; Li et al., 2023b), previous VQA benchmarks listed in Tab. 2 are not tailored for evaluating MLLMs with open-from output, since they require an exact match between the model prediction and the target word or phrase. The qualitative examples of multimodal comprehension is provided in Appendix. F.

**Text-to-image Generation.** We evaluate the text-to-image generation on MS-COCO (Chen et al., 2015) and Flickr30K (Young et al., 2014) and compute the pair-wise CLIP similarity score as the evaluation metric following GILL (Koh et al., 2023b). As shown in Tab. 10, images generated by our SEED-LLaMA from textual descriptions show higher similarity with the ground-truth images. The results demonstrate that SEED-LLaMA generates images that are highly correlated with text prompts via a frozen SD-UNet. We show qualitative examples of text-to-image generation in Appendix. F.

### 4.1.2 EMERGENT ABILITY

**Multi-turn In-context Multimodal Generation.** As shown in Fig. 1 and Fig. 6, given multimodal instructions including images and open-form texts from a user, our SEED-LLaMA can respond with synthesized image (*e.g.*, a dog in front of the Golden Gate Bridge), sequentially generated images (*e.g.*, a cartoon cat in different scenes), instruction-followed image (*e.g.*, a closer look-up of a cherry

blossom), various forms of texts via creation and real-world knowledge (*e.g.*, a story, a poem and flower identification). The results illustrate the impressive capability of SEED-LLaMA in reasoning and generating long-context multimodal content.

**Compositional Image Generation.** As shown in Fig. 7, our SEED-LLaMA can realize a variety of zero-shot compositional image generation as below,

- Stylized Image Generation. SEED-LLaMA can take a text prompt and a style reference image as inputs and produce an output image that adheres to both the style and text prompt.
- Image Blending. SEED-LLaMA can take two images as inputs and generate an image that blends the visual components of the input images.
- Multimodal Composition. SEED-LLaMA can take an image prompt and a text prompt as inputs and generate a composite image that combines the multimodal inputs.
- In-context Generation. SEED-LLaMA can take images, their textual references, and text prompts as inputs and generate context-related images.

## 4.2 ABLATION STUDY

**Generation Embedding.** The generation embedding of SEED is aligned with the image embedding of unCLIP-SD, and can be decoded to realistic images with the unCLIP-SD-UNet. We also train a visual tokenizer SEED[text] through aligning the generation embeddings with the text embeddings (77 tokens) of SD (Rombach et al., 2022) conditioned on texts. As shown in Tab. 10, the similarity between the reconstructed images of SEED[text] and original images drop heavily. The semantic representations of texts can not fully preserve the rich visual information of images. The visual comparison of the the reconstructed images between SEED[text] and SEED are provided in Appendix. A.

**Causal Visual Codes vs. Bilateral Visual Codes.** We train a Causal Q-Former to convert 2D features produced by the ViT encoder into a sequence of causal semantic embeddings, which are further discretized as causal visual codes. To verify whether the causal visual codes are necessary for compatibility with LLM, we train a visual tokenizer SEED[Bi], which produces bilateral visual codes from a pre-trained Q-Former with bilateral self-attention. We then pre-train SEED[Bi]-LLM* and SEED-LLM* on image-text pairs and evaluate the text-to-image generation on COCO test set. Given 5000 captions of COCO, SEED[Bi]-LLM only generates 2134 images successfully while SEED-LLM* generates 4997 images (Failure cases occur when the model predicts a number of visual tokens not equal to 32). The results demonstrate that the non-causal codes lead to highly unstable model performance since they contradict with the left-to-right autoregressive mechanism of LLM.

**SEED-LLaMA Pretraining.** We first train SEED-LLaMA using LoRA tuning, and then merge the parameters of LoRA with the original LLM and fine-tune all parameters except for the embedding layer. To explore whether fully fine-tuning helps, we evaluate the performance of the model before and after fully fine-tuning on image captioning and text-to-image generation, with evaluation metric CIDEr and clip similarity score. Tab. 3 shows that fully fine-tuning the LoRA tuned model enhances model's capability for both image comprehension and generation.

Table 3: Evaluation of image captioning and text-to-image generation on COCO test set.

| Pretraining | Captioning | Generation |
|---|---|---|
| LoRA | 124.5 | 68.87 |
| LoRA + Fully | **125.0** | **69.07** |

## 5 CONCLUSION

We present SEED, a discrete image tokenizer, designed based on the premise that visual tokens compatible with LLMs should capture high-level semantics while being generated with 1D causal dependency. SEED enables LLMs to be trained with multimodal data following the original recipe of text (*i.e.*, next-word prediction), which is mature and scalable. We further present SEED-LLaMA via multimodal pretraining and instruction tuning on the interleaved visual and textual data with SEED tokenizer. SEED-LLaMA not only exhibits remarkable performance across multimodal comprehension and image generation tasks, but also demonstrates extensive compositional emergent abilities. We hope that SEED would draw increased attention to visual tokenizers. A more rational visual tokenizer could substantially reduce the complexity of multimodal LLM training.

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

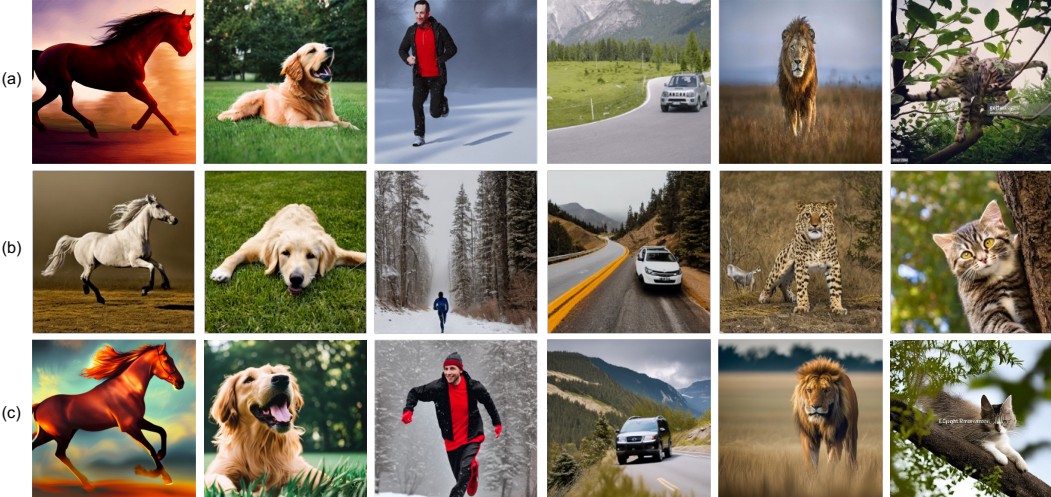

Figure 8: (a) Input image. (b) Reconstruction images of SEED[text] tokenizer, which is aligned with the feature space of a SD conditioned on text embeddings. (c) Reconstruction images of SEED tokenizer, which is aligned with the feature space of a SD conditioned on image embedding.

## A    SEED TOKENIZER

The generation embedding of SEED is aligned with the image embedding of unCLIP (Ramesh et al., 2022) SD, and can be decoded to realistic images with the unCLIP-SD-UNet. We also train a visual tokenizer SEED[text] through aligning the generation embeddings with the text embeddings (77 tokens) of SD (Rombach et al., 2022), and the generation embeddings can be decoded to images with the SD-UNet. The visual comparison of the the reconstructed images between SEED[text] and SEED are shown in Fig. 8. We can observe that compared with SEED[text], the images reconstructed by SEED can better preserve the visual information of the original images.

## B    PRETRAINING

### B.1    PRETRAINING DATA

As shown in Tab. 4, we utilize diverse categories of datasets as pretraining data, which can be summarized as follows.

**Image-text Pairs.** We use the image-text pairs from CC3M (Sharma et al., 2018), Unsplash (Luke Chesser, 2023), LAION-COCO (Christoph et al., 2022) and MS-COCO (Chen et al., 2015). We filtered the samples in these datasets based on image resolution, aspect ratio, and visual-textual similarity. We randomly place images or text at the forefront, in order to achieve the generation of captions based on images and vice versa.

**Video-text Pairs.** We use a large-scale dataset WebVid-10M (Bain et al., 2021) containing videos and captions. We implemented heuristic rules to exclude extraneous metadata, such as the resolution of the original video and camera parameters. We sample four frames of each video for training. Each frame is discretized into 32 codes. The input sequence of a video is  the token of frame 1 </IMG>  the token of frame 2 </IMG>  the token of frame 3 </IMG>  the token of frame 4 </IMG>.

**Interleaved Image and Text.** We use publicly available MMC4 (Zhu et al., 2023b) and OBELISC (Laurençon et al., 2023) datasets, which were extracted and thoroughly filtered from Common Crawl. Specifically, we employ the MMC4-core split, consisting of 7.3 million samples, and the complete OBELISC dataset, containing 141 million samples. For documents in MMC4, we create a sequence of length 1024 and randomly shuffle the order of images and their corresponding texts (those with the

Table 4: Description of pretraining datasets of SEED-LLaMA.

| Dataset Name | Dataset Description |
|---|---|
| COCO Caption (Chen et al., 2015) | 0.5M image-text pairs with human-written captions. Specifically, Karpathy train split is used. |
| CC3M (Sharma et al., 2018) | 3.3M image-text pairs from the web. |
| Unsplash (Luke Chesser, 2023) | 4.8M image-text pairs, in which images are composed of high-quality Unsplash photos. |
| LAION-COCO (Christoph et al., 2022) | 600M image-text pairs, where the caption is generated by the BLIP (Li et al., 2022). |
| MMC4 (Zhu et al., 2023b) | 101M image-interleaved documents collected from Common Crawl. We use the mmc4-core split which is consist of 7.3M documents. We randomly shuffle the order of images and their corresponding text (those with the highest CLIP score). |
| OBELISC (Laurençon et al., 2023) | 141M image-interleaved documents collected from Common Crawl. |
| WebVid (Bain et al., 2021) | 8M video-text pairs, we have implemented heuristic rules to exclude extraneous metadata,such as the resolution of the original video and camera parameters. |

Table 5: Summary of pretraining hyperparameters of SEED-LLaMA.

| Configuration | SEED 8B | SEED 14B |
|---|---|---|
| Vision encoder | EVA-CLIP | |
| LLM | Vicuna-7B | LLaMA2-Chat-13B |
| Training Strategy | LoRA + Fully fine-tuning | |
| Peak learning rate | 1.5e-4 | |
| Warmup ratio | 0.03 | |
| LR schedule | Cosine decay | |
| Optimizer | AdamW | |
| Optimizer hyper-parameters | $\beta_1, \beta_2, \epsilon = 0.9, 0.98$, le-6 | |
| Image resolution | $224 \times 224$ | |
| Weight decay | 0.05 | |
| Iterations | 30k + 10k | |
| Data | (MS-COCO, CC3M, Unsplash), LAION-COCO, OBELISC, MMC4, WebVid | |
| Sequence length per dataset | 160, 128, 1024, 1024, 200 | |
| Batch size per dataset | 146, 180, 26, 26, 116 | 46, 56, 8, 8, 36 |
| Sample ratio per dataset | 4.5%, 54.5%, 9.1%, 27.3%, 4.5% | |

highest CLIP score). As for OBELISC, we generate a sequence of length 1024 based on the order of data in the dataset.

## B.2 PRETRAINING HYPERPARAMETERS

We report the detailed pretraining hyperparameters of SEED-LLaMA in Tab. 5.

## C  INSTRUCTION TUNING

We summarize the datasets and their prompts for supervised instruction tuning of SEED-LLaMA in Tab. 6 and Tab. 7. Note that MagicBrush (Zhang et al., 2023a) contains both the single-turn and multi-turn scenarios, and we only use the single-turn for multimodal prompt image generation.

## D  EVALUATION

### D.1  BENCHMARKS

In order to assess the multimodal comprehension and image generation ability of SEED-LLaMA, we evaluate SEED-LLaMA on 10 benchmarks as shown in Tab. 8. For the evaluation of image generation, we adopt the CLIP-ViT-L/14 to calculate the CLIP score between the ground-truth image and the generated image. When evaluating SEED-Bench, we adhere to the official guidelines, selecting the option with the highest log likelihood as the response for each multi-choice question (MCQ). For the evaluation on video tasks, we uniformly sample 4 frames for MSVDQA and MSRVTTQA, and 8 frames for NExTQA. For the other tasks, we follow the evaluation procedures in prior works (Li et al., 2023c; Sun et al., 2023b) and either submit the results to the official server (VQAv2, VizWiz) or assess them using the official code ourselves.

### D.2  PROMPT TEMPLATES

We summarize the prompt templates used for evaluating SEED-LLaMA in Tab. 9. As the pre-trained SEED-LLaMA with size of 8B and 14B adopt different LLM (Vicuna-7B and Llama2-chat-13B), their prompts differ accordingly.

### D.3  EVALUATION OF IMAGE RECONSTRUCTION

We further evaluate image reconstruction on COCO and Flickr30K dataset. SEED first discretizes input images into causal codes (32 tokens) and obtain generation embedding (1 token), which are fed into the unCLIP-SD-UNet for reconstruction. We follow GILL (Koh et al., 2023a) to compute the CLIP similarity score as the metric to evaluate the semantic consistency. As shown in Tab. 10, compared with the upper bound unCLIP-SD, SEED only slightly drops performance.

## E  QUALITATIVE COMPARISON

For the qualitative comparison of multi-turn in-context image and text generation, as shown in Fig. 9, we can observe that our SEED-LLaMA exhibits exceptional emergent ability in generating long-context multimodal content, while Emu Sun et al. (2023b) and Next-GPT Wu et al. (2023) that employ continuous representations to unify comprehension and generation within a LLM show less competitive performance. They generate images with incorrect semantics or context, or can not follow instruction accurately in a multi-turn dialogue.

## F  QUALITATIVE CASES

More examples of multi-turn in-context multimodal generation and compositional image generation are shown in Fig. 10 and Fig. 11. Note that generating images with multimodal prompt is not an emergent ability since SEED-LLaMA is fine-tuned on corresponding paired data such as Instruct-Pix2Pix (Brooks et al., 2023). We showcase qualitative examples of text-to-image generation by SEED-LLaMA in Fig. 12. Given various textual descriptions, our SEED-LLaMA can generate realistic images that aligns with the text prompts. We further provide qualitative examples of multi-modal comprehension by SEED-LLaMA in Fig. 13, Fig. 14 and Fig. 15. SEED-LLaMA can realize in-context multi-image understanding, real-world knowledge grounding, complex reasoning, story creation and video understanding.

Table 6: Description of datasets in the instruction tuning of SEED-LLaMA.

| Task | Dataset Name | Dataset Description | Type |
|---|---|---|---|
| Text-to-Image Generation | JourneyDB (Pan et al., 2023) | It contains 4429K Midjourney images, with text prompt, image caption, and QA pairs. | Single-turn |
| | DiffusionDB (Wang et al., 2022) | It contains 14 million images generated by Stable Diffusion using prompts by real users. | Single-turn |
| | LAION-Aesthetics | It contains several collections of subsets from LAION 5B with high visual quality. | Single-turn |
| | VIST (Huang et al., 2016) | It contains photos in 20K sequences, aligned to both caption and story language. | Multi-turn |
| Multimodal Prompt Image Generation | Instructpix2pix (Brooks et al., 2023) | It contains text editing instructions and the corresponding images, with 454K samples. | Single-turn |
| | MagicBrush (Zhang et al., 2023a) | It contains 10K manually annotated triplets (source image, instruction, target image). | Single-turn |
| Image Conversation | LLaVA (Liu et al., 2023b) | We use 58K multi-turn conversations between an assistant and a person. | Multi-turn |
| | SVIT (Zhao et al., 2023) | It contains conversations, complex reasoning, referring QA and detailed image description. | Multi-turn |
| | LLaVAR (Zhang et al., 2023c) | It contains 16K multi-turn conversations, each with QA pairs for text-rich images. | Multi-turn |
| Multi-Image Understanding | GSD (Li et al., 2023a) | It contains 141K pairs of images with text describing the differences. | Single-turn |
| Image Captioning | VSR (Liu et al., 2023a) | It contains texts describing the spatial relations in the image, with 7K training samples. | Single-turn |
| | COCO Caption (Chen et al., 2015) | It contains image-text pairs with human-written captions, with 82K training samples. | Single-turn |
| | TextCaps (Sidorov et al., 2020) | It requires the model to comprehend and reason the text in images, with 21K training samples. | Single-turn |
| Image QA | VQAv2 (Goyal et al., 2017) | A dataset for open-ended image question answering, with 82K training samples. | Single-turn |
| | OKVQA (Marino et al., 2019) | It contains questions that require outside knowledge to answer, with 9K training samples. | Single-turn |
| | A-OKVQA (Schwenk et al., 2022) | It is a successor of OKVQA containing more challenging questions, with 17K training samples. | Single-turn |
| | GQA (Hudson & Manning, 2019) | It contains questions for image understanding and reasoning, with 30K training samples. | Single-turn |
| | VizWiz (Gurari et al., 2018) | It contains visual questions asked by people who are blind, with 20K training samples. | Single-turn |
| | TextVQA (Singh et al., 2019) | It contains questions that require models to read text in the image, with 800K training samples. | Single-turn |
| | OCR-VQA (Mishra et al., 2019) | It contains questions that requires reasoning about text to answer, with 173K training samples. | Single-turn |
| Video Conversation | Video-ChatGPT (Maaz et al., 2023) | It contains of 100K video-instruction pairs via manual and semi-automated pipeline. | Single-turn |
| Video QA | ActivityNet (Caba Heilbron et al., 2015) | It contains 200 different types of activities from YouTube, with 10K training videos. | Single-turn |
| | Next-QA (Xiao et al., 2021) | It contains 52K QA pairs of videos grouped into causal, temporal and descriptive questions. | Single-turn |
| | MSVD (Chen & Dolan, 2011) | It contains videos from YouTube with descriptions, containing 1.2K training samples. | Single-turn |
| | MSR-VTT (Xu et al., 2016) | It contains videos from YouTube with descriptions, containing 19K training samples. | Single-turn |
| | iVQA (Yang et al., 2021) | It is a video QA dataset with mitigated language biases, containing 6K training samples. | Single-turn |

Table 7: Details of prompt templates used in supervised instruction tuning of SEED-LLaMA.

| Type | Prompt |
|---|---|
| Text-to-Image Generation | USER: {caption} Please generation an image.\nASSISTANT: {image} |
| Multimodal Prompt Image Generation | USER: {image1} {instruction} Please generation an image. \nASSISTANT: {image2} |
| Image Conversation | USER: {image} {question}\nASSISTANT: {answer} |
| Multi-Image Understanding | USER: This is the first image. {image1} This is the second image. {image2} {question}\nASSISTANT: {answer} |
| Image Captioning | USER: {image} Please provide an accurate and concisedescription of the given image.\nASSISTANT: {caption} |
| Image QA | USER: {image} {question} Please provide an accurate answer consisting of only one word or phrase.\nASSISTANT: {answer} |
| Video Conversation | USER: {video} {question}\nASSISTANT: {answer} |
| Video QA | USER: {video} {question} Please provide an accurate answer consisting of only one word or phrase.\nASSISTANT: {answer} |

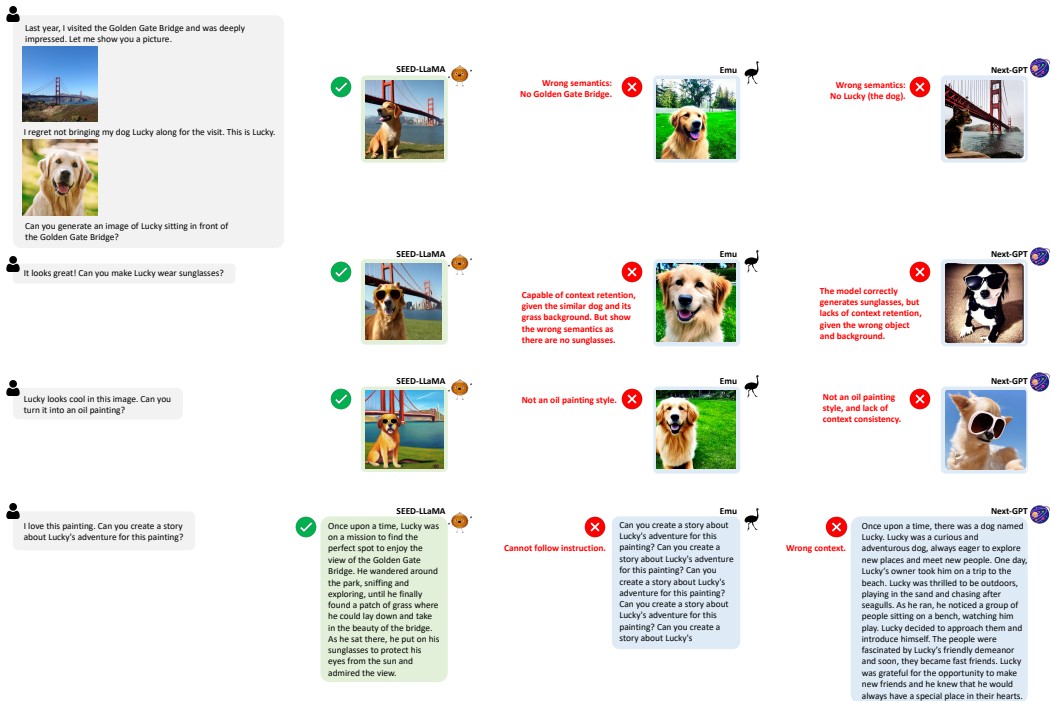

Figure 9: Comparison of multi-turn in-context image and text generation between SEED-LLaMA and MLLMs that unify comprehension and generation.

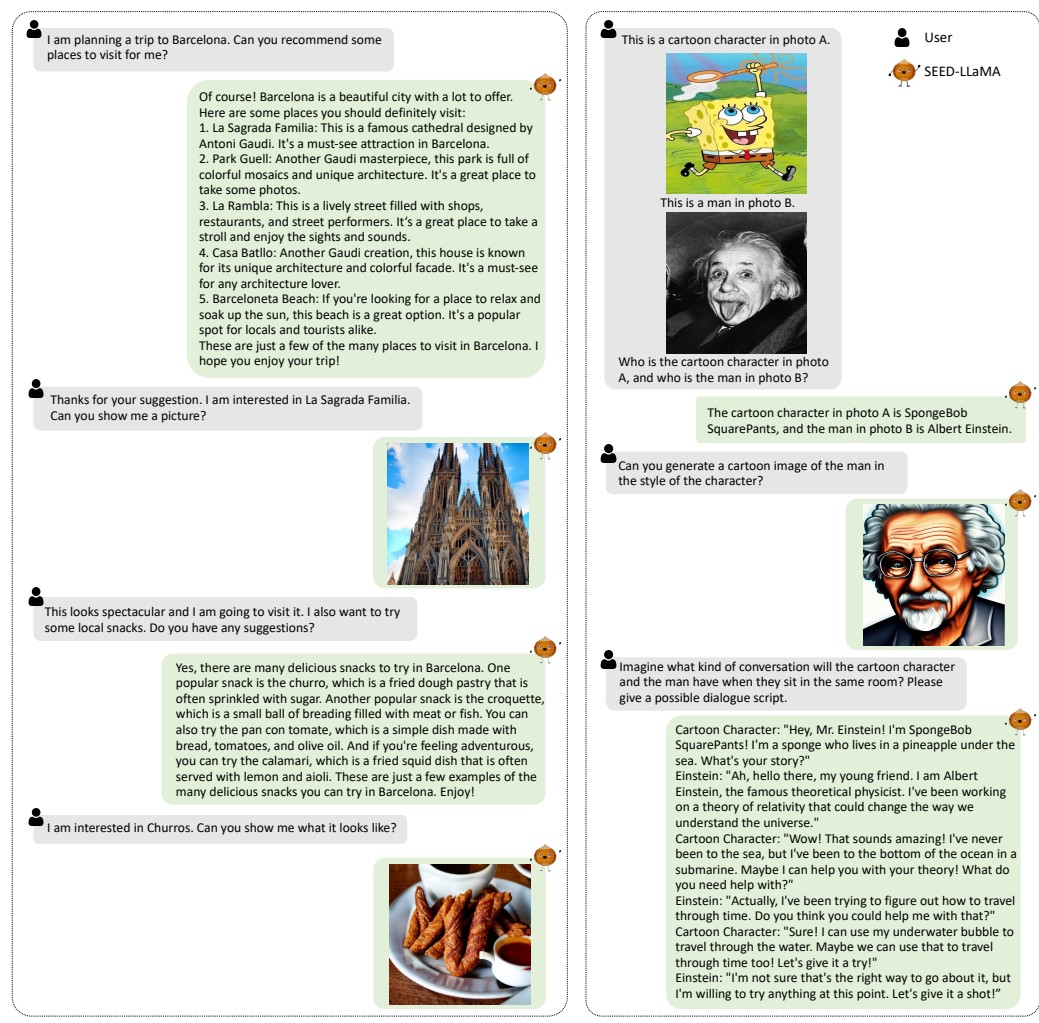

Figure 10: Qualitative examples of multi-turn in-context image and text generation by SEED-LLaMA given multimodal instructions.

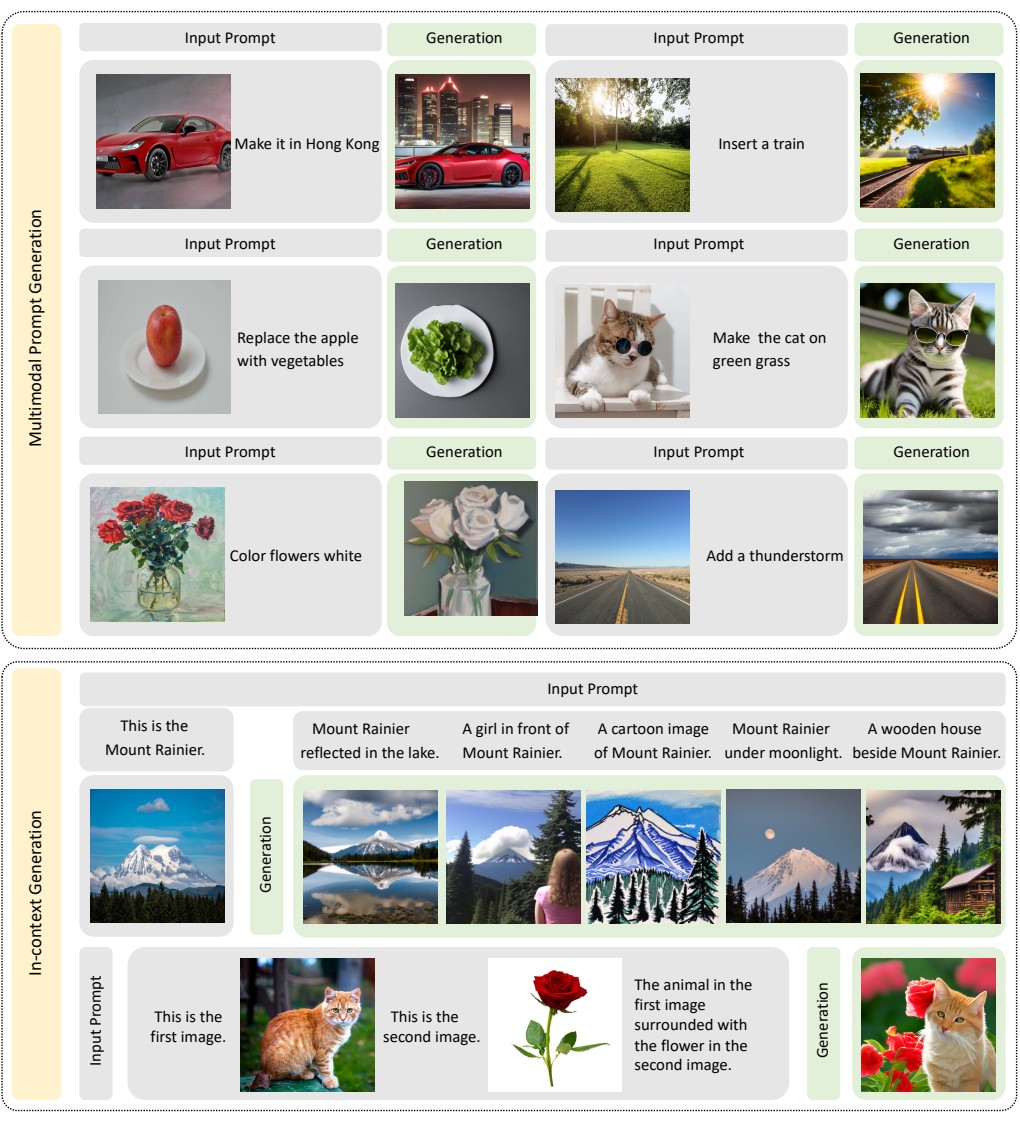

Figure 11: Qualitative examples of compositional image generation by SEED-LLaMA.

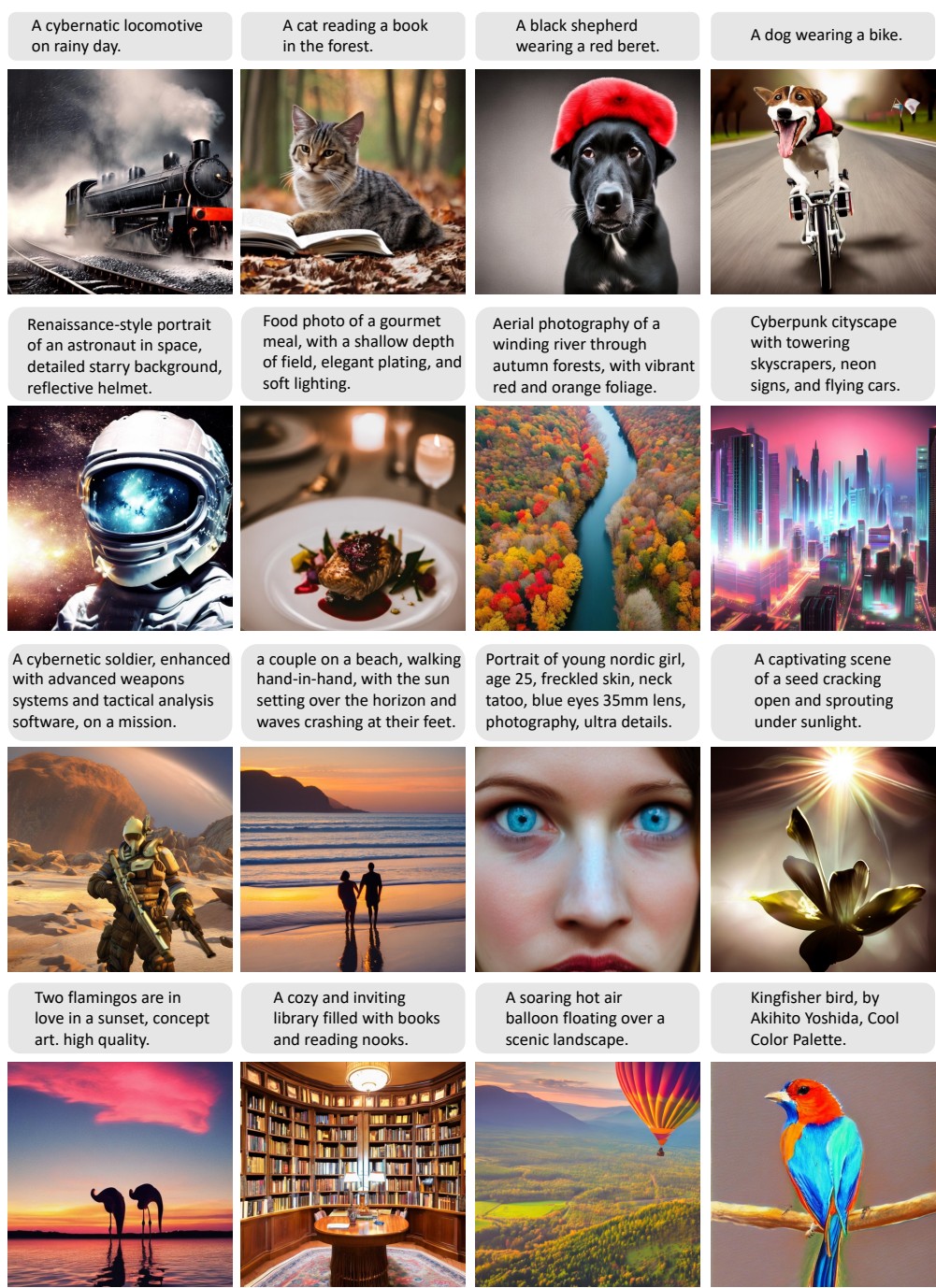

Figure 12: Qualitative examples of text-to-image generation by SEED-LLaMA.

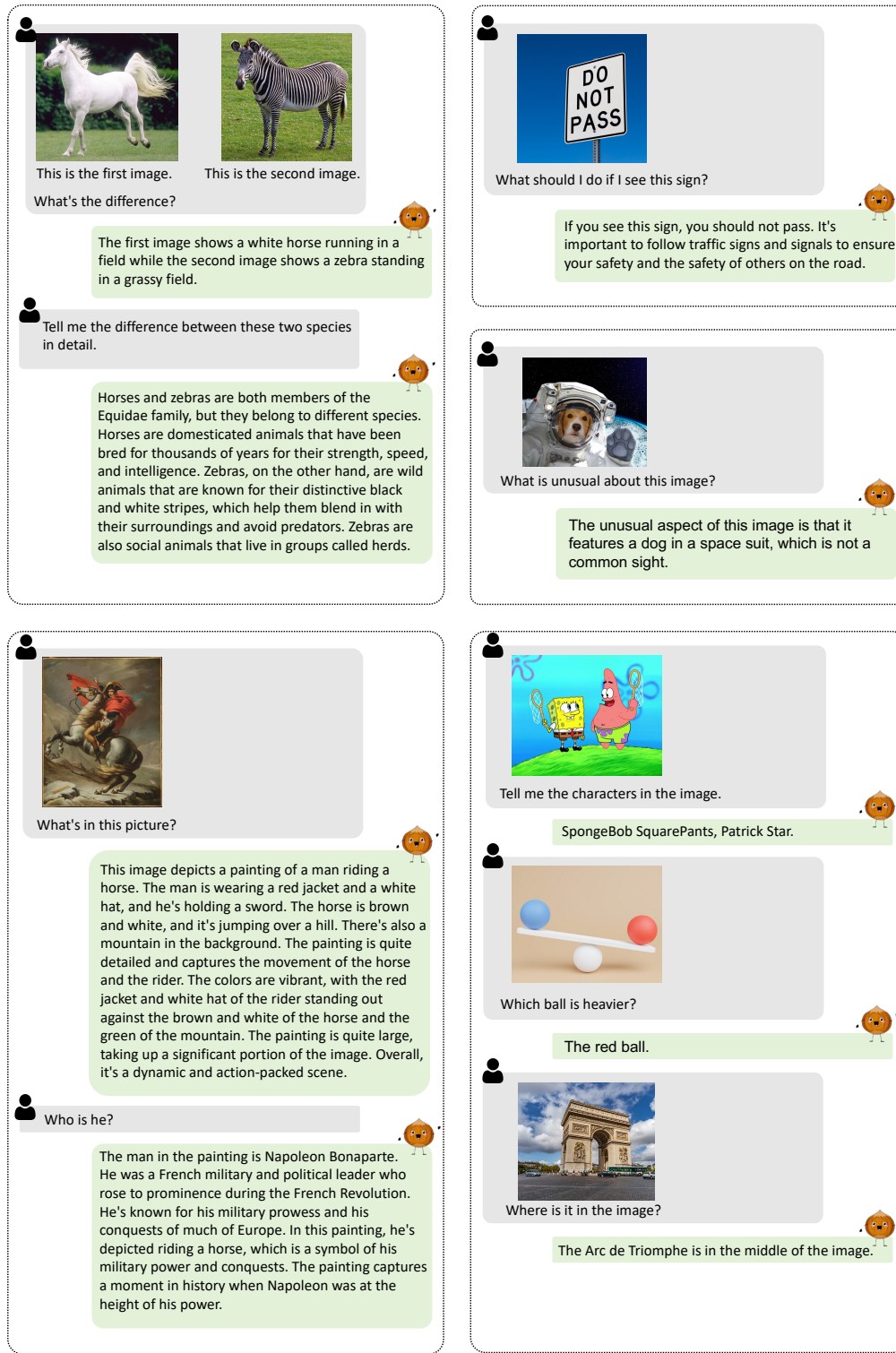

Figure 13: Qualitative examples of multimodal comprehension by SEED-LLaMA.

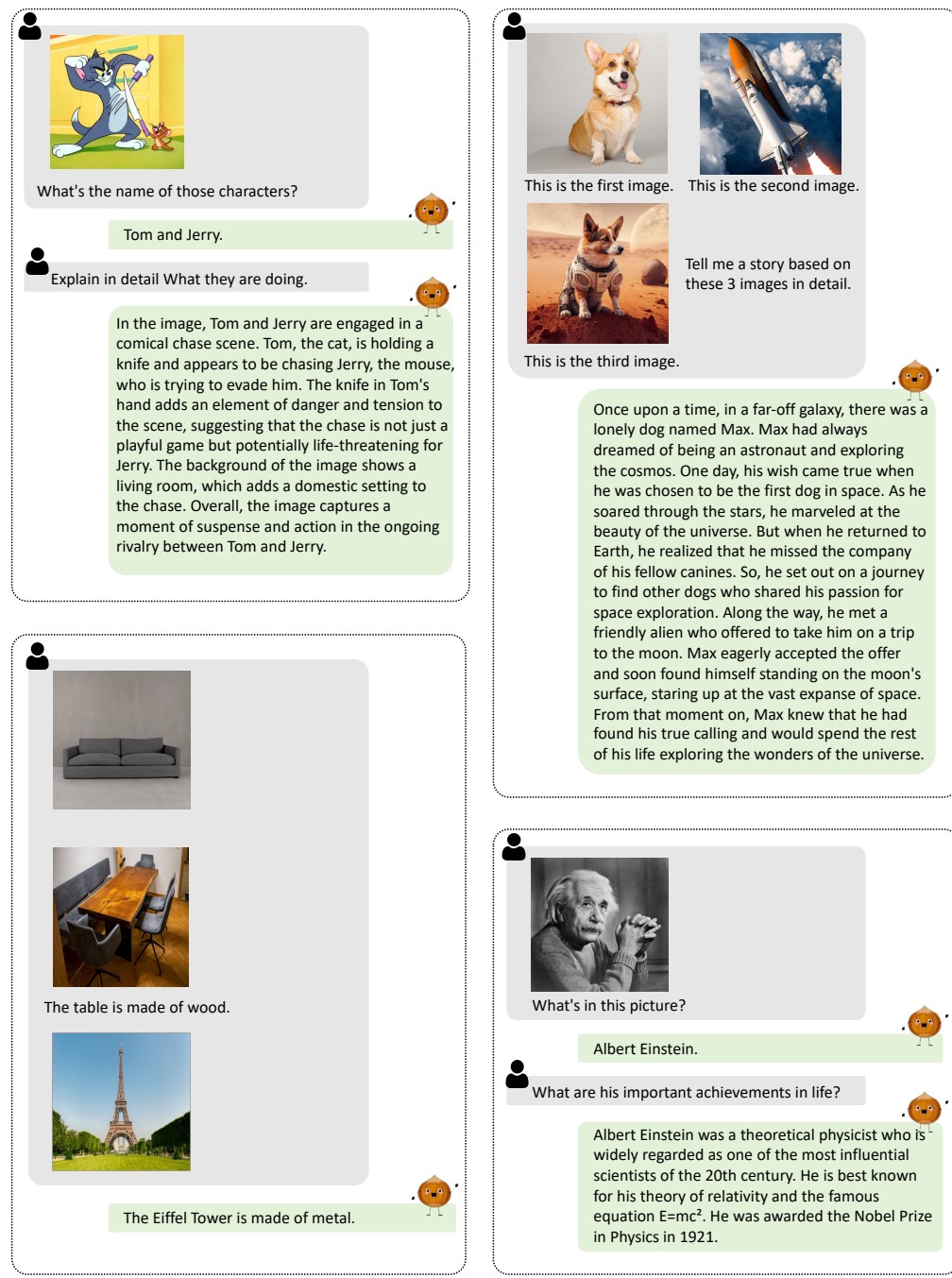

Figure 14: Qualitative examples of multimodal comprehension by SEED-LLaMA.

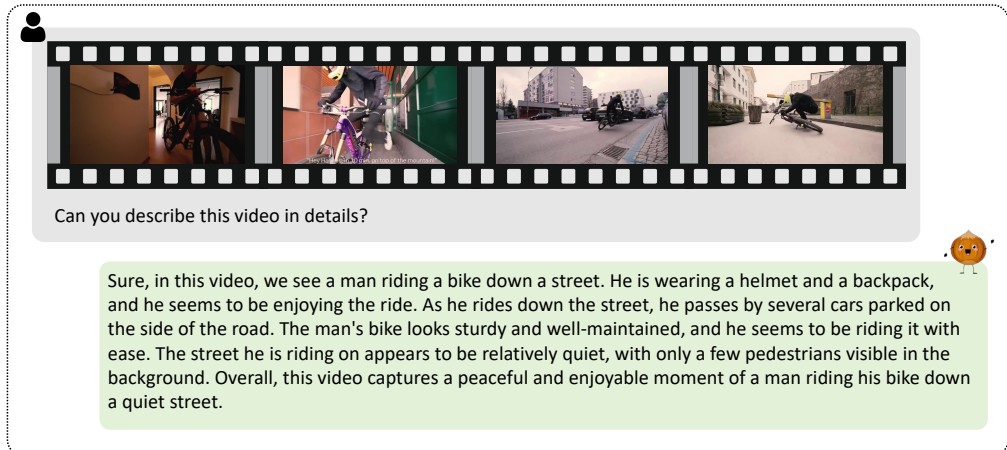

Can you describe this video in details?

Sure, in this video, we see a man riding a bike down a street. He is wearing a helmet and a backpack, and he seems to be enjoying the ride. As he rides down the street, he passes by several cars parked on the side of the road. The man's bike looks sturdy and well-maintained, and he seems to be riding it with ease. The street he is riding on appears to be relatively quiet, with only a few pedestrians visible in the background. Overall, this video captures a peaceful and enjoyable moment of a man riding his bike down a quiet street.

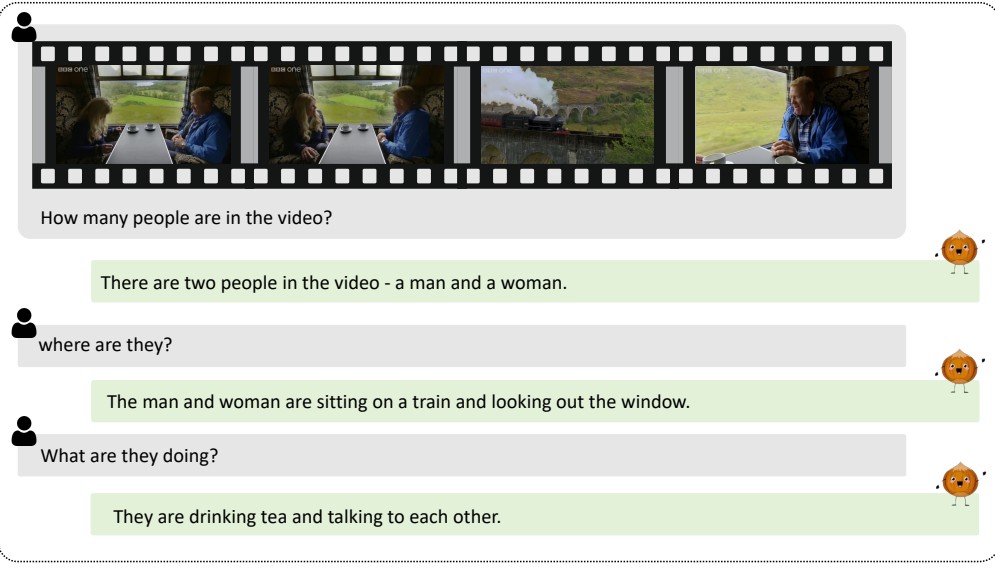

How many people are in the video?

There are two people in the video - a man and a woman.

where are they?

The man and woman are sitting on a train and looking out the window.

What are they doing?

They are drinking tea and talking to each other.

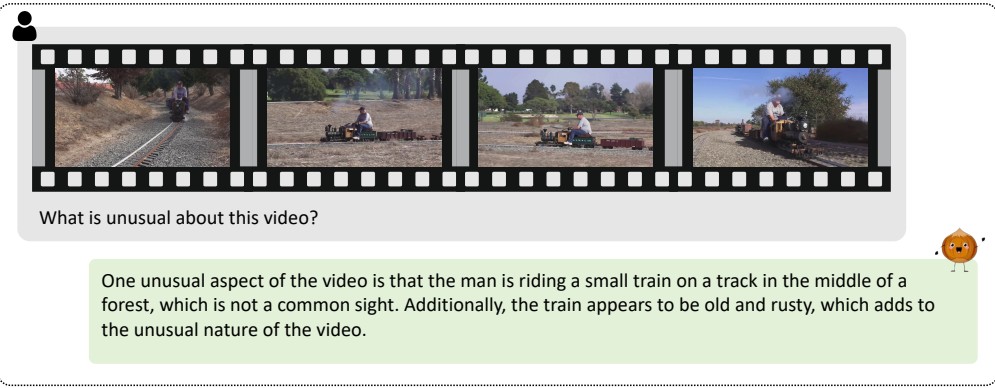

What is unusual about this video?

One unusual aspect of the video is that the man is riding a small train on a track in the middle of a forest, which is not a common sight. Additionally, the train appears to be old and rusty, which adds to the unusual nature of the video.

Figure 15: Qualitative examples of multimodal comprehension by SEED-LLaMA.

Table 8: Summary of the evaluation benchmarks.

| | Dataset | Task | Split | Metric |
|---|---|---|---|---|
| Image | COCO(Lin et al., 2014) | Text-to-Image Generation | Karpathy test | CLIP score (↑) |
| | Flickr30K (Young et al., 2014) | Text-to-Image Generation | test | CLIP score (↑) |
| | COCO Caption (Chen et al., 2015) | Scene Description | test | CIDEr (↑) |
| | VQAv2 (Goyal et al., 2017) | Scene Understanding QA | test-dev | VQA acc. (↑) |
| | OKVQA (Marino et al., 2019) | External Knowledge QA | val | VQA acc. (↑) |
| | VizWiz (Gurari et al., 2018) | Scene Understanding QA | test-dev | VQA acc. (↑) |
| | SEED-Bench (Li et al., 2023b) | Comprehensive QA | dim 1-9 | MCQ acc. (↑) |
| Video | MSVDQA (Chen & Dolan, 2011) | Event Understanding QA | test | Top-1 acc. (↑) |
| | MSRVTTQA (Xu et al., 2016) | Event Understanding QA | test | Top-1 acc. (↑) |
| | NExTQA (Yang et al., 2021) | Temporal/Causal QA | test | WUPS (↑) |

Table 9: Summary of the prompting template for evaluating SEED-LLaMA.

| Model | Type | Template |
|---|---|---|
| SEED-LLaMA 8B | Image Captioning | {image} |
| | Image QA | {image}USER: {question} Please provide an accurate answer consisting of only one word or phrase.\nASSISTANT: |
| | Video QA | {video}USER: {question} Please provide an accurate answer consisting of only one word or phrase.\nASSISTANT: |
| SEED-LLaMA 14B | Image Caption | {image} |
| | Image QA | {image} Please provide an accurate answer consisting of only one word or phrase based on the image.\n Question:{question} \n Answer: |
| | Video QA | {video} Please provide an accurate answer consisting of only one word or phrase based on the video.\n Question:{question}\n Answer: |
| SEED-LLaMA-I 8B & 14B | Image Caption | USER: {image}Please provide an accurate and concise description of the given image.\nASSISTANT: |
| | Image QA | USER: {image}{question} Please provide an accurate answer consisting of only one word or phrase.\nASSISTANT: |
| | Video QA | USER: {video}{question} Please provide an accurate answer consisting of only one word or phrase.\nASSISTANT: |

Table 10: Evaluation of Image Generation.

| Model | COCO | Flickr30K |
|---|---|---|
| *Image-to-image* | | |
| unCLIP (Ramesh et al., 2022) SD | **79.30** | **79.55** |
| SEED[text] | 68.23 | 65.22 |
| SEED | 77.35 | 76.52 |
| *Text-to-image* | | |
| GILL (Koh et al., 2023b) | 67.45 | 65.16 |
| Emu (Sun et al., 2023b) | 66.46 | 64.82 |
| SEED-LLaMA | 69.07 | 65.54 |
| SEED-LLaMA-I | **70.68** | **66.55** |

