# OpenReview forum: "Making LLaMA SEE and Draw with SEED Tokenizer"
_ICLR.cc/2024/Conference — ICLR 2024 poster_

### Official Review · Reviewer_TFPg · 2023-10-30

**Soundness:** 3 good
**Presentation:** 3 good
**Contribution:** 3 good
**Rating:** 8
**Confidence:** 4

**Summary:**

The paper presents a method (SEED) to augment large language models (LLMs) with the capability of processing and generating visual data, i.e., images. The core contribution of SEED is a quantized tokenizer that learns to encode images into discrete visual tokens which can be again decoded using a pre-trained generative model. Once the tokenizer is trained, a LLM is trained and fine-tuned on interleaved image-text data such that the LLM can both process and generate the visual tokens which make it applicable to a variety of vision language tasks.

**Strengths:**

- SEED is well motivated in learning a 1D token representation that better aligns with the auto-regressive generative process of LLMs.
- Architectural choices are reasonable and have been validated with ablation studies to the most extend, i.e., text vs visual embedding reconstruction, embedding vs. discrete code, causal vs. bilateral visual codes, full fine-tuning after LoRA, instruction fine-tuning.
- The quantitative evaluation convinces in either surpassing or being competitive in image-text, video-text tasks as well as generative image-to-image and text-to-image tasks.
- The qualitative results showcase some interesting multi-modal capabilities of SEED, including compositional and in-context image generation.
- Publishing both code and checkpoints of large-scale models enables future research and empowers the open-source community.

**Weaknesses:**

- Some architectural design choices are missing an explanation.
    - In general, it would help the clarity of the paper if all loss functions would be written out.
    - What is the reason behind using two different image encoders, one for encoding the input to the causal q-former (BLIP-2 ViT) and one for the image generation conditioning (unCLIP-SD vision encoder)? This requires loading more weights into memory during training so an explanation is needed. Can we use the unCLIP-SD vision encoder for both cases?
    - How important is the contrastive loss between the vision and text embeddings? An ablation could justify it's inclusion.
    - Why was the original VQ codebook loss replaced by a simple reconstruction loss with cosine similarity? How are collapsing codebooks avoided? Are there any stop-gradient operation in the loss for the codes?
- The arguments and ablation for using causal vs. bilateral visual codes is not convincing. In general, previous work on VQ models have demonstrated that transformers can learn complex and even low-level dependencies of non-causal codes. Enforcing a causal mask in the q-former restricts the information flow to fully utilize the tokens efficiently and effectively. The argument made in Sec. 4.3 is that the LLM struggles with generating the correct sequence length for images which should always be 32. It is surprising that this happens because the start token for images (as in Fig. 4) should be a clear signal to the LLM that 32 image tokens follow. In practice however, it would be straightforward to enforce the generation of 32 image tokens after the start token is observed by restricting the possible output tokens. How do these two models compare when the number of image tokens is enforced to be always 32?
- It is not clear how videos are being processed. Are individual frames used to train the tokenizer or are multiple frames passed to the causal q-former? If multiple frames are passed, how do you adjust the reconstruction loss for the generative embedding (1 embedding per frame from unCLIP-SD vs. one embedding per video from your tokenizer)? Do you simply append the encoding of multiple frames when passing videos to the LLM?

**Questions:**

- How did you decide on using 32 tokens per image and 8192 codes?
- Can you confirm that you are using a start and end token for images as shown in Fig. 4? Do you use the same start/end tokens for both images and videos? This information should be included in the paper.

Suggestions:
- It would help the read flow to already mention the codebook size in section 3.1.2 instead of only in 3.2.1 (i.e., 8192 visual codes).
- When Table 2 is first discussed in Sec. 4.1, it is unclear what $\text{SEED}^{\text{text}}$ refers to. A short description and reference to the ablation in Sec. 4.3. would better facilitate immediate understanding for the reader. Similarly for the "I" suffix referencing the instruction-tuned model.

---

> ### Author Response · Authors · 2023-11-18
>
> **W1-1: Write loss functions.**
>
> **Response:** Thank you for the advice. We have revised the method section and included all loss functions in our revised manuscript.
>
> **W1-2: Why use two different image encoders.**
>
> **Response:** We use two different image encoders in order to ease training by leveraging the pre-trained knowledge of BLIP-2 and unCLIP-SD. Specifically, we initialize the Causal Q-Former from the pre-trained BLIP-2 Q-Former that is compatible with BLIP-2 ViT, so we use the ViT encoder from BLIP-2 for image encoding. And we would like to decode images using the off-the-shelf SD (frozen), so we adopt the image encoder (disposable after training) from unCLIP-SD to provide the learning target for our generation embeddings. After training, the generation embeddings can be used in place of the original conditional embedding of SD-UNet.
>
> Since the image encoder of unCLIP-SD is frozen during training, only the parameters need to be loaded onto GPU, occupying 2944MiB memory (12.7% of total memory 23260MiB for batch size 512).
>
> It is theoretically feasible to use only the unCLIP-SD encoder - Causal Q-Former requires random initialization. We will leave it for future work.
>
>
> **W1-3: The importance of contrastive loss.**
>
> **Response:** It is critical since the Causal Q-Former is trained using only the contrastive loss. Specifically, the training of visual tokenizer consists of two stages: 1) Causal Q-Former, and 2) VQ Codebook and MLP.
>
> We have attempted to employ a single-stage approach to optimize all components but the codebook was hard to be optimized and eventually collapsed. This may be due to the continuous evolution of the reconstruction targets of discrete codes (i.e., the features of the Causal Q-Former).
>
> To conclude, for stable training, the Causal Q-Former needs to be optimized individually whereas contrastive learning acts as the sole objective and thus cannot be removed.
>
>
>
> **W1-4: The training objectives of VQ.**
>
> **Response:** Sorry for missing details of training VQ codebook. We adopt (1) the original VQ codebook loss, (2) a reconstruction loss between reconstructed causal embeddings and causal embeddings of Causal Q-Former, (3) a reconstruction loss between generation embedding and image embedding of unCLIP-SD to optimize VQ codebook. To avoid codebook collapse, in addition to the two-stage training method mentioned above, we follow BEiT v2[1] to reduce the dimension of codebook embeddings to 32D. We use the stop-gradient operation in the original VQ codebook loss. We have revised the method section with detailed introduction.
>
> > [1] BEiT v2: Masked Image Modeling with Vector-Quantized Visual Tokenizers
>
>
> **W2: The performance of bilateral visual codes when the number of tokens is enforced to 32.**
>
> **Response:** Such an operation is applicable only when the model predicts equal to or more than 32 tokens. We have attempted to truncate the visual tokens in such cases, however, there are still 2 failure cases given 5000 captions in COCO test set. The results of two models when the number of image tokens is enforced to be always 32 are listed as below. We can observe that the model with causal visual codes achieve higher similarity between generated images and groundtruth images (i2i), and between generated images and captions (i2t) than the model with bilateral visual codes.
>
>
> | | CLIP score (i2i) | CLIP score (i2t) |
> | -------- | -------- | -------- |
> |  Bilateral    | 70.08     | 23.06     |
> |  Causal    | 70.30     | 23.57     |
>
>
>
> **W3: How to process videos.**
>
> **Response:** We train the SEED tokenizer with only image-text pairs (i.e., the Causal Q-Former and generation embeddings are only trained on image data), and only use video-text data for training SEED-LLaMA.
>
> When we train SEED-LLaMA on videos, we sample four frames from each video. Each frame is discretized into 32 codes. The input sequence of a video is `<IMG> the token of frame 1 </IMG> <IMG> the token of frame 2 </IMG> <IMG> the token of frame 3 </IMG> <IMG> the token of frame 4 </IMG>`.
>
>
>
> **Q1: Why 32 tokens and 8192 codes?**
>
> **Response:** The numbers are selected based on the practice in existing works, i.e., 32 queries in BLIP-2[1] and a visual vocabulary of 8192 codes in BEIT V2[2].
>
> > [1] BLIP-2: Bootstrapping Language-Image Pre-training with Frozen Image Encoders and Large Language Models
> > [2] BEIT V2: Masked Image Modeling with Vector-Quantized Visual Tokenizers
>
>
> **Q2: Start and end token for images and videos.**
>
> **Response:** We use a start and end token for each image as shown in Fig.4. The same start and end token is used for both images and video frames (mentioned in the response to **W3**). We have included this information in section 3.2.1.
>
> **Q3: Suggestions.**
>
> **Response:** Thank you a lot for the useful suggestions. We have mentioned the codebook size in section 3.1.2, and added corresponding description and reference in the revised manuscript.

---

> > ### Comment · Reviewer_TFPg · 2023-11-20
> >
> > I would like to thank the authors for their detailed reply and the updates to the paper.
> > Below I have some follow-up comments.
> >
> > > **W1-1: Write loss functions.**
> >
> > This makes the manuscript much clearer.
> >
> > > **W1-3: The importance of contrastive loss.**
> > > Response: It is critical since the Causal Q-Former is trained using only the contrastive loss. Specifically, the training of visual tokenizer consists of two stages: 1) Causal Q-Former, and 2) VQ Codebook and MLP.
> > > We have attempted to employ a single-stage approach to optimize all components but the codebook was hard to be optimized and eventually collapsed. This may be due to the continuous evolution of the reconstruction targets of discrete codes (i.e., the features of the Causal Q-Former).
> > > To conclude, for stable training, the Causal Q-Former needs to be optimized individually whereas contrastive learning acts as the sole objective and thus cannot be removed.
> >
> > Since the Causal Q-Former is trained in isolation with the contrastive loss, it is a bit surprising that the loss is only applied on the last token (if I understand correctly). In theory, the Causal Q-Former could learn to ignore the first 31 tokens and embed all information from the image into the last token to optimize the loss.
> >
> > BLIP-2 informed the choice of using 32 tokens, but BLIP-2 also use all tokens in their loss functions. This choice could have been justified a bit better, e.g., ablate number of tokens with 1 as the extreme case, or use the same contrastive loss as BLIP-2 which takes all pairwise similarities and then select the highest one.
> >
> > > **W2: The performance of bilateral visual codes when the number of tokens is enforced to 32.**
> > > Response: Such an operation is applicable only when the model predicts equal to or more than 32 tokens.
> >
> > I disagree. One could simply sample 32 tokens from the subset of image tokens once the start token is observed, i.e., take the softmax only over all image tokens and then sample, repeating this procedure exactly 32 times. This ensures that the next 32 tokens are only image tokens and the "image end token" could even be automatically appended as well. Therefore, it should come down to the actual performance difference instead of structural errors and failure cases in the sequence.
> >
> > > We can observe that the model with causal visual codes achieve higher similarity between generated images and groundtruth images (i2i), and between generated images and captions (i2t) than the model with bilateral visual codes.
> >
> > It is good to see that it makes a difference, although I would argue it is a marginal one. It is also not clear which setting (e.g. dataset) is evaluated here. I still believe that the quite small benefit of creating causal tokens does not justify the focus it is given throughout the paper. It would be important to include these results in the paper, ideally in Sec. 4.3.
> >
> > > When we train SEED-LLaMA on videos, we sample four frames from each video. Each frame is discretized into 32 codes. The input sequence of a video is ```<IMG> the token of frame 1 </IMG> <IMG> the token of frame 2 </IMG> <IMG> the token of frame 3 </IMG> <IMG> the token of frame 4 </IMG>```.
> >
> > The details about how the prompt looks like for video data should be included in the supplementary.

---

> > > ### Author Response · Authors · 2023-11-22
> > >
> > > **W1-3: Loss applied on the last token of Casual Q-Former.**
> > > **Response:** Your understanding of training Causal Q-Former is correct. The contrastive loss is applied on the final embedding of the Causal Q-Former. However, if the Causal Q-Former ignores the first 31 tokens, when calculating self-attention for the final token, the key and value of the first 31 tokens will become semantically void, which makes it difficult to optimize the final token. In practice, our experiments demonstrate that the first 31 tokens from Causal Q-Former are **semantically meaningful** as shown below, where different output query tokens are used for text-image retrieval on the COCO test set. We can further observe that the retrieval performance of the subsequent tokens is enhanced, as they effectively aggregate information from the preceding tokens.
> > >
> > > |  | 1-th | 4-th |12-th |20-th | 28-th |32-th |
> > > | -------- | -------- | -------- |-------- |-------- | -------- |-------- |
> > > | R@mean    |   65.8   |  70.2  |     77.8 | 80.5|82.0|82.5
> > >
> > > BLIP-2 uses all tokens in their loss functions through taking all pairwise similarities and then selecting the highest one, since it **does not aim to impose dependency among query tokens**. By contrast, we aim to establish causal dependency among query tokens by employing causal self-attention among tokens and optimizing the final token using contrastive loss. The above experiment has validated that information gradually aggregates from preceding tokens to subsequent ones, thus demonstrating the effectiveness of applying contrastive learning to the final token of Causal Q-Former, which also saves memory during training.
> > >
> > > Thank you for the suggestion. In future work, we will investigate the supervision of other tokens during training Causal Q-Former and explore its influence on the performance of SEED-LLaMA, considering that pretraining can not be finished within this rebuttal period.
> > >
> > >
> > > **W2: The performance of bilateral visual codes when the number of tokens is enforced to 32.**
> > > **Response:** We have followed your suggestion by sampling 32 tokens from the subset of image tokens once the start token is observed, ensuring the generation of 32 image tokens for the ablated model with bilateral visual codes. We evaluate the text-to-image generation on COCO and Flickr30K test set and compute the pair-wise CLIP similarity score following GILL[1] as below.
> > >
> > > |  | COCO (i2i) | COCO (i2t) |Flickr30K (i2i) | Flickr30K (i2t) |
> > > | -------- | -------- | -------- | -------- | -------- |
> > > | Bilateral    | 70.09     | 23.07     |65.04     | 22.01     |
> > > | Causal    | 70.30    | 23.57    |66.66    | 23.14    |
> > >
> > > We can observe that the model with causal visual codes achieve higher similarity between generated images and groundtruth images (i2i), and between generated images and captions (i2t) than the model with bilateral visual codes on both dataests. We want to highlight that the improvement in CLIP similarity socre is not marginal, as GILL[1] points out in their paper ''Our model outperforms SD, improving CLIP Similarity from 0.598 to 0.612''.
> > >
> > > We further evaluate the performance of zero-shot visual question answering tasks on VQAv2, OKVQA and GQA as below, to investigate the influence of bilateral or causal codes for multimodal **comprehension**. The evaluation metric is accuracy. We use the same prompt `{image} Question: {question}\nBrief answer:` for both models.
> > > |  | VQAv2 | OKVQA |GQA |
> > > | -------- | -------- | -------- | -------- |
> > > | Bilateral    | 23.52    | 15.07    |18.34     |
> > > | Causal    | 39.40    | 21.54    |27.22    |
> > >
> > >
> > > We can observe that the model using bilateral visual codes significantly drops the performance on multimodal comprehension tasks. When only considering understanding tasks (text as the only output) for training, bilateral visual codes have a better theoretical performance due to the rich semantics. However, in SEED-LLaMA, where the understanding (text as output) and generation (image as output) tasks are unified, the autoregressive Transformer fails to be properly optimized when using bilateral codes as it struggles with next-visual-token prediction. As a result, it achieves inferior performance on understanding tasks. The same phenomenon is reported in our concurrent work[2] (see its Appendix B.2)
> > >
> > > > [1] Generating Images with Multimodal Language Models
> > >
> > > > [2] Unified Language-Vision Pretraining in LLM with Dynamic Discrete Visual Tokenization
> > >
> > > We have included the results and the discussions in Sec. 4.3.
> > >
> > >
> > > **Suggestion.**
> > >
> > > **Response:** Thank you for the suggestion. We have included the details of the prompts for video data in the Appendix B.1.

---

> > > > ### Comment · Reviewer_TFPg · 2023-11-23
> > > >
> > > > Thank you for running the additional experiments in the short time and providing extensive explanations.
> > > >
> > > > Both the individual token evaluations and the causal vs biliteral comparisons are very interesting and highly relevant.
> > > > They provide good support for your design choices and clarify my concerns.
> > > >
> > > > I think the ablation of individual tokens with text-image retrieval on the COCO is interesting enough that it warrants inclusion in the appendix.
> > > >
> > > > Regarding the updated causal vs biliteral paragraph, I just have some minor suggestions:
> > > > - mention the table in the text
> > > > - add the experimental setting for the generation column for instance like this: Generation **(i2i)**
> > > >
> > > >
> > > > Overall, the rebuttal has improved the paper and strengthened my evaluation of acceptance. I think my initial score was already fair in this regard, and I will advocate for this decision in the reviewer discussion.

---

### Official Review · Reviewer_i2L6 · 2023-10-31

**Soundness:** 2 fair
**Presentation:** 3 good
**Contribution:** 2 fair
**Rating:** 6
**Confidence:** 5

**Summary:**

This work introduces an image tokenizer, which is capable of discretizing images into a series of tokens. These image tokens are transformed by Q-Former into pseudo-causal tokens that can serve as the input for Large Language Models (LLMs), most importantly, they can also act as the target. This allows the model to unify both visual understanding and generation tasks.

**Strengths:**

This paper is the first (at least to my knowledge) to use an image tokenizer to unify visual understanding and generation tasks in LLM, providing a feasible pipeline.

**Weaknesses:**

1. The authors claim that the use of q-former can establish a causal dependency, which I find questionable. This is because the attention in the visual encoder stage is bidirectional, leading to potential information leakage.

2. Regarding the visual understanding task results shown in Table 3, why does SEED-LLaMA-I (14B) perform no better (or nearly the same) as SEED-LLaMA-I (8B) on some Image-Text Tasks? Does the proposed method not yield much gain on larger models, or has it already reached saturation?

3. In Table 2, SEED-LLaMA-I achieves good results. However, I believe that CLIP similarity does not effectively reflect the quality of generation. Fréchet Inception Distance (FID) is a widely accepted better evaluation method, but unfortunately, this paper does not provide it.

4. Regarding Section 4.3 (Causal Visual Codes vs. Bilateral Visual Codes), the authors mention that some mode collapse may occur for generation tasks, but what about understanding tasks?

**Questions:**

See weakness.

---

> ### Author Response · Authors · 2023-11-18
>
> **W1: The information leakage by bidirectional attention in ViT.**
>
> **Response:** The bidirectional attention in ViT is among 2D patches while the unidirectional causal attention in Causal Q-Former is among 1D queries. It signifies that the information of the current query is derived based on the context provided by previous queries. It is worth noting that the causal queries capture **sparse-to-dense** semantics from the **entire image** rather than only a part of an image, as demonstrated in Fig. 9 of Appendix. So it is ok for each query to see all ViT features via cross-attention. Also, according to the information flow (image$\rightarrow$frozen ViT, w/ init queries$\rightarrow$Q-Former), the semantics of $k$-th query would not be leaked to $\{k-1\}$-th query via ViT.
>
>
> **W2: SEED-LLaMA-I (14B) does not surpass SEED-LLaMA-I (8B) on some Image-Text tasks**.
>
> **Response:** These traditional VQA benchmarks (VQAv2, OKVQA, VizWiz) fail to properly reflect the model's multimodal capabilities since they require an exact match between the model prediction and the ground truth, as pointed out by recent works[2,3,4]. However, empowered by LLM, the model predictions that have been penalized may even be better than the ground truth with a single word or phrase.
>
> We can observe that SEED-LLaMA-I (14B) outperforms SEED-LLaMA-I (8B) on SEED-Bench, which is specifically designed for evaluating MLLMs with open-form output. Such results have indicated the scalability of SEED-LLaMA.
>
> The same phenomenon is also reported in IDEFICS (TODO CITE), where  IDEFICS-I (80B) achieves worse performance than IDEFICS-I (9B) on VQAv2, OKVQA, VizWiz but much better on SEED-Bench. Check out the results below.
>
>
> |  | VQAv2 | OKVQA | VizWiz | SEED-Bench|
> | -------- | -------- | -------- | -------- | -------- |
> |SEED-LLaMA-I (8B)    | 66.2     | 45.9    | 55.1     | 51.5     |
> | SEED-LLaMA-I (14B)    | 63.4     | 43.2      | 49.4     | 53.7   |
> | IDEFICS-I (9B)    | 65.8    | 46.1     | 41.2    | 41.2    |
> | IDEFICS-I (80B)     | 37.4     | 36.9    | 26.2     | 53.2    |
>
>
> > [1] Obelics: An open web-scale filtered dataset of interleaved image-text documents
> > [2] SEED-Bench: Benchmarking Multimodal LLMs with Generative Comprehension
> > [3] MMBench: Is Your Multi-modal Model an All-around Player?
> > [4] MME: A Comprehensive Evaluation Benchmark for Multimodal Large Language Models
>
>
> **W3: Need FID for generation evaluation.**
>
> **Response:** We would like to clarify that FID is mostly adopted to measure the **quality** of generated images by comparing the distribution between a set of generated images and a set of real images. However, to properly evaluate our SEED tokenizer, we focus more on the **semantical consistency** between the original input image and the reconstructed image. FID fails to meet the requirements, and CLIP similarity is more suitable in such a case, which is also adopted by GILL[1]. Furthermore, SEED aims to better tokenize images and uses the off-the-shelf SD-UNet decoder. Quality is generally determined by the decoder and is not specifically optimized in our work.
> We further evaluate the text-to-image generation on MS-COCO with FID as the evluation metric as below. We can observe that our model achieves competitve performance.
>
> |  | FID |
> | -------- | -------- |
> | GILL    | 12.20    |
> | Emu    | 11.66     |
> | SEED-LLaMA     | 12.33     |
>
>
> > [1] Generating Images with Multimodal Language Models
>
>
> **W4: "Causal Visual Codes vs. Bilateral Visual Codes" in understanding tasks.**
>
> **Response:** We evaluate the performance of zero-shot visual question answering tasks on VQAv2, OKVQA and GQA (accuracy as the metric), and image captioning on COCO (CIDEr as the metric) as below,
>
> |  | VQAv2 | OKVQA |GQA | COCO
> | -------- | -------- | -------- | -------- |-------- |
> | Bilateral    | 23.52    | 15.07    |18.34     | 118.8    |
> | Causal    | 39.40    | 21.54    |27.22    | 120.1 |
>
> We can observe that the model with bilateral visual codes significantly drops the performance in understanding tasks. When only considering understanding tasks (text as the only output) for training, bilateral visual codes have a better theoretical performance due to the rich semantics. However, in SEED-LLaMA, where the understanding (text as output) and generation (image as output) tasks are unified, the autoregressive Transformer fails to be properly optimized when using bilateral codes as it struggles with next-visual-token prediction for generation tasks. As a result, it achieves inferior performance on understanding tasks as well.

---

> > ### Comment · Reviewer_i2L6 · 2023-11-22
> >
> > Thanks for the authors’ efforts in rebuttal.  The response solves my concerns, and as a result, I have decided to increase my score.

---

### Official Review · Reviewer_y9Sh · 2023-11-06

**Soundness:** 3 good
**Presentation:** 2 fair
**Contribution:** 3 good
**Rating:** 5
**Confidence:** 3

**Summary:**

Making the language model to see the words is one of the key research direction. This paper present new image tokenization on this direction. Specifically, unlike prior attempts that uses simple 2d style image tokenization (usually VQ-VAE), this paper propose SEED, which makes image embedding to be left-to-right 1d tokenization similar to the text while keeping semantic meaning of images but discarding low-level information. This paper claim that capturing too low-level information hider the performance of LLMs to effectively perform multimodal comprehension.

**Strengths:**

(1) Unifying vision and text representation is one of the hot research topic.
(2) The assumption behind the proposal is reasonable.
(3) The paper is generally well-written.

**Weaknesses:**

(1) The methodology is not religiously explained and is not self-contained. Especially, section 3.1 is hard to follow. There are no equation, and it is hard to track which components are trained on which objective function.

(2) In section 4.1, they compared SEED tokenization on image-text retrieval. As described in the paper, SEED generally outperform BLIP-2, in some case BLIP-2 exceed the proposed method. However, there are no explanation on this point. Similar criticism can be applied for the analysis on Table 3.

(3) Regarding the Figure 7, I'm afraid the actual prompt is hard to imagine.

(4) The proposed method contains several components, including image encoder, codebook, text encoder, and generation module. However, the importance of all components is less discussed, making me hard to access the importance of the specific choice of each component.

(5) I'm afraid that I found a statement in the introduction is not fully validated. "Moreover, we empirically found that the dominant tokenizer VQ-VAE (Van Den Oord et al., 2017) in existing works captures too low-level information for LLMs to effectively perform multimodal comprehension task". Could you please clarify again which results support the above statement?

(6) While the paper motivate to learn good 1D representation is key to incorporate visual information into pre-trained LLMs, but less discuss on why we should make the representation discrete rather than continuous. Table1 also seems to show that the continuous representation is generally better than discrete representation.

**Questions:**

See weakness section

---

> ### Author Response · Authors · 2023-11-18
>
> **W1: Not religiously explained and self-contained. Sec. 3.1 is hard to follow. No equation.**
>
> **Response:** Thank you for the suggestion. We designed the SEED tokenizer with two premises: (1) image tokens should capture high-level semantics, and (2) image tokens should be produced under 1D causal dependence. The architecture of SEED meets the above requirements with the sub-modules of a pre-trained CLIP-ViT encoder, a Causal Q-former, a VQ codebook, and a De-Tokenizer which is composed of an MLP and the pre-trained SD-UNet.
>
> Intuitively, the pre-trained **CLIP-ViT encoder** captures **high-level semantics**, the **causal Q-former** produces **1D causal embeddings**, which are subsequently discretized into **1D causal IDs** using the VQ codebook. The produced visual IDs enable LLM to perform scalable multimodal autoregression under its original training recipe, i.e., next-word prediction. The **de-tokenizer** is used to **reconstruct images**, employed in case the follow-up multimodal LLM (i.e., SEED-LLaMA) predicts image token IDs and needs to render them into pixels further. Note that, to pursue efficient training, we directly adopt the pre-trained and frozen **SD-UNet decoder**, and train a lightweight **MLP** to transform the visual codes to the conditional embedding space of UNet.
>
> During SEED training, only Causal Q-Former, VQ Codebook, and the MLP are tunable. There are two training stages. First, we optimize the causal Q-Former by image-text contrastive loss in the form of InfoNCE. Then, we jointly train the VQ codebook and the MLP. Besides the original VQ objectives, we adopt dual reconstruction, i.e., the reconstruction of continuous causal embeddings (to avoid code collapse) and the reconstruction of conditional generation embeddings (to be compatible with the off-the-shelf SD-UNet). The **auxiliary text encoder and image encoder (disposable after training)** are derived from BLIP-2 and unCLIP-SD respectively, providing learning targets.
>
> We have carefully revised Sec. 3.1 in blue, adding equations and objective functions for each learnable component (Enq.1 for Causal Q-former, Enq.2 for VQ codebook, and Enq.3 for Multi-layer perceptron). Please find more details in our revised manuscript.
>
> **W2: Lack of analysis of Table 1 and Table 3.**
>
> **Response:** Thank you for the suggestion.
>
> (1) *Regarding Table 1:* We would like to first clarify that our goal is not to surpass BLIP-2, since it is a comprehension-only model whereas our SEED is tailored for unifying comprehension and generation in one framework. It is worth noting that to achieve the goal of unification, we would inevitably sacrifice the performance of comprehension to achieve autoregressive multimodal generation. Specifically, the 1D causal dependence (unidirectional attention) and discrete tokens are both designed to accommodate generation but are harmful to discriminative image representation learning. Here we took BLIP-2 as a baseline for comparison since we adopted its pre-trained ViT and Qformer as initialization (mentioned in Sec. 3). As expected, our causal embeddings (before VQ) achieve better results in terms of image-to-text retrieval due to more training data, while falling short in text-to-image retrieval that relies on more discriminative image features. Generally speaking, with our elaborate training, the Causal Qformer achieves competitive results even with unidirectional attention.
>
> (2) *Regarding Table 3:* The table benchmarks the ability of multimodal comprehension. Although we achieve multimodal generation at the expense of discriminative representations (discussed above), we still obtain competitive comprehension results compared to SOTAs and much better performance than the ones that also unify comprehension and generation (such as CM3Leon and Emu). SEED-LLaMA generally complies with the scaling law, i.e., more parameters lead to better results as shown in SEED-Bench (a benchmark tailored for MLLMs). And we found that the traditional benchmarks (e.g., VQAv2) are no longer sufficient to properly evaluate the MLLMs due to even better predictions than the ground-truth.
>
> More explanations can be found in our revised manuscript.

---

> > ### Author Response · Authors · 2023-11-18
> >
> > **W3: The prompt of Figure 7.**
> >
> > **Response:** Sorry for the confusion. The prompt we used (now Figure. 11 in the Appendix) is as below.
> >
> > 1. Stylized image generation: `<IMG> the token of reference image </IMG> Generate an image of [CAPTION HERE] in the same style.`
> > 2. Image blending:  `<IMG> the token of image1 </IMG> <IMG> the token of image2 </IMG> Please blend the two images.`
> > 3. Multimodal composition:  `<IMG> the token of input image </IMG> [INSTRUCTION HERE].`
> > 4. In-context generation:  `<IMG> the token of image1 </IMG> This is the first image. <IMG> the token of image2 </IMG> This is the second image. [INSTRUCTION HERE].`
> >
> > Actually, SEED-LLaMA can process open-ended prompts, not limited to the above templates.
> >
> > **W4: The importance of each component.**
> >
> > **Response:** Thank you for the suggestion. The design intuition, function, and training objectives of the main modules in the SEED tokenizer have been discussed in the response to **W1**. We for the first time introduce such an architecture where every component is essential as it has its own functionality. But the inner model might be replaced with advanced ones with similar functionality in the future. For instance, the ViT encoder and UNet decoder might be replaced with stronger ones, e.g., ViT-G and SD-XL. Check out the brief introduction of each component below.
> >
> > *Tokenization:*
> > - ViT encoder: It aims to encode high-level visual features from raw image pixels, which serves as the inputs of the Causal Q-Former, and is kept frozen.
> > - Causal Q-Former: It aims to convert 2D raster-ordered features (16×16 tokens) from the image encoder into a sequence of causal embeddings (32 tokens) as the inputs of the codebook.
> > - VQ Codebook: It aims to discretize the causal embeddings from Causal Q-Former to quantized visual codes (32 tokens) with causal dependency, so that LLM is able to perform scalable multimodal autoregression under its original training recipe, *i.e.*, next-word prediction, with SEED tokens.
> > - Text encoder (only for training): It aims to encode text features so that we can adopt contrastive learning to optimize Causal Q-Former, which maximizes the similarity between the final causal embedding of Causal Q-Former and text features of the corresponding caption, and minimizes the final causal embedding and text features of other captions in a batch.
> >
> > *De-tokenization:*
> > - MLP+UNet: It aims to decode generation embedding from discrete codes, which reconstructs the image embedding of the unCLIP-SD, so that the generation embedding can be used to generate realistic images given the off-the-shelf SD-UNet.
> > - Image encoder (only for training): It is derived from unCLIP-SD and encodes the image embedding to provide the reconstruction targets for training the generation module.
> >
> > **W5: Evidence of VQ-VAE’s inferior performance in multimodal understanding tasks.**
> >
> > **Response:** We conduct a pilot experiment of two baseline tokenizers, VQ-VAE trained by reconstructing image pixels, and BEiT v2[1] trained by reconstructing high-level ViT features. We respectively align discrete representations of VQ-VAE and  BEiT v2 with OPT-2.7B model on CC3M dataset, and evaluate the performance with zero-shot image captioning on COCO. VQ-VAE achieves CIDEr 34.0 while Beit v2 achieves 42.0. The experimental results demonstrate that a visual tokenizer, which captures low-level image details, is not effective for multimodal comprehension.
> >
> > > [1] BEiT v2: Masked Image Modeling with Vector-Quantized Visual Tokenizers
> >
> > The statement can also be supported by a concurrent work CM3Leon[2], which utilizes a pre-trained VQ-VAE as the image tokenizer to perform multimodal autoregression. However, it yields sub-optimal performance in multimodal comprehension tasks (e.g., CIDEr 61.6 vs. ours 126.9 on COCO image captioning). We have added such detailed discussion in the revised manuscript.
> >
> > > [2] Scaling Autoregressive Multi-Modal Models: Pretraining and Instruction Tuning

---

> > > ### Author Response · Authors · 2023-11-18
> > >
> > > **W6: Why discrete?**
> > >
> > > **Response:** Thank you for the question. We would like to clarify that we have never argued for the absolute superiority of discrete representations over continuous ones. In fact, this remains *an open question* to date. However, there are several natural advantages of discrete visual representations in unifying multimodal comprehension and generation.
> > >
> > > **A. Simplicity.** Representing images as a sequence of discrete tokens is naturally compatible with the autoregressive training objective of LLMs, so that LLM can be trained using only the next-”word” prediction loss (cross-entropy) without specific adaptation for multimodal data. In such a way, you can directly borrow many useful techniques (e.g., the efficient codebase) from LLMs. Otherwise, if adopting continuous embeddings, you need to adopt cross-entropy for text and L2-regression for images. You need careful hyperparameter tuning, code refactorization, and more to ensure efficient and effective training.
> > >
> > > **B. Efficiency.** Representing images as discrete tokens allows us to pre-process visual inputs into IDs, which substantially conserve computational resources and storage space (interleaved image-text dataset OBELISC occupies only 6% of the original dataset storage space when using 32 IDs to represent one image). During training, only the parameters of LLM are loaded onto the GPU. We do not need visual encoders or decoders which require additional memory to enable full gradient graphs, as is done in those continuous solutions.
> > >
> > > **B. Scalability.** With discrete visual tokens, LLM can perform scalable multimodal autoregression via large-scale training, which is empirically verified as the key to the exceptional performance and emergent capabilities of MLLM. Scalability is born based on the simplicity and efficiency mentioned above. Simplicity enables the reuse of existing mature techniques, e.g., the easy-to-use framework for large-scale distributed training, and the well-proven autoregression loss. Efficiency allows more tokens to be learned within the same computational resources. As demonstrated in Fig. 8 of Appendix in our revised manuscript, we can observe that our SEED-LLaMA exhibits exceptional emergent ability in multi-turn in-context image and text generation, which can be attributed to scalable multimodal autoregression in large-scale interleaved image-text data. By contrast, Emu[1] and Next-GPT[2] that employ continuous representations to unify comprehension and generation within a LLM show less competitive performance. They generate images with incorrect semantics or context, or can not follow instruction accurately in a multi-turn dialogue.
> > > > [1] Generative Pretraining in Multimodality
> > >
> > > > [2] NExT-GPT: Any-to-Any Multimodal LLM

---

> > > > ### Author Response · Authors · 2023-11-23
> > > >
> > > > Thank you for your time and effort in reviewing our paper. We would like to remind that the rebuttal period is approaching its end. If you have any other comments or questions, please let us know.

---

### Author Response · Authors · 2023-11-22
**A kind reminder regarding our response**

We thank you for your time and effort in reviewing our paper. We have responded to all comments in the rebuttal. We would like to remind that the rebuttal period is approaching its end. If you have any other comments or questions, please let us know.

Thank you for your attention.

---

### Meta-Review · Area_Chair_PZWV · 2023-12-06

**Metareview:**

1x A, 1x BA, and 1x BR. This paper proposes to quantize tokens to augment LLM for image generation. The reviewers agree on the (1) important topic, (2) reasonable motivation, (3) novel idea, and (4) convincing results. Although some concerns appeared, e.g., (1) unclear technical details, and (2) missing in-depth ablation studies., the rebuttal has addressed them. The AC leans to accept this submission.

**Justification For Why Not Higher Score:**

N/A

**Justification For Why Not Lower Score:**

N/A

---

### Decision · Program_Chairs · 2024-01-16

Accept (poster)